# NeuralPlane: Structured 3D Reconstruction in Planar Primitives with Neural Fields

**Hanqiao Ye**[1,2]**, Yuzhou Liu**[1,2]**, Yangdong Liu**[2]**, Shuhan Shen**[1,2]*

[1]School of Artificial Intelligence, University of Chinese Academy of Sciences
[2]Institute of Automation, Chinese Academy of Sciences

## Abstract

3D maps assembled from planar primitives are compact and expressive in representing man-made environments. In this paper, we present **NeuralPlane**, a novel approach that explores **neural** fields for multi-view 3D **plane** reconstruction. Our method is centered upon the core idea of distilling geometric and semantic cues from inconsistent 2D plane observations into a unified 3D neural representation, which unlocks the full leverage of plane attributes. It is accomplished through several key designs, including: 1) a monocular module that generates geometrically smooth and semantically meaningful segments known as 2D plane observations, 2) a plane-guided training procedure that implicitly learns accurate 3D geometry from the multi-view plane observations, and 3) a self-supervised feature field termed Neural Coplanarity Field that enables the modeling of scene semantics alongside the geometry. Without relying on prior plane annotations, our method achieves high-fidelity reconstruction comprising planar primitives that are not only crisp but also well-aligned with the semantic content. Comprehensive experiments on ScanNetv2 and ScanNet++ demonstrate the superiority of our method in both geometry and semantics. Project page: https://neuralplane.github.io/

## 1 Introduction

While inferring dense geometry such as volumetric grids and meshes from 2D images has been extensively studied (Schönberger et al., 2016; Murez et al., 2020; Wu et al., 2023b), there is a growing realization that reconstructs *sparse 3D maps composed of self-contained primitives* (Xue et al., 2024; Li et al., 2024; Kluger et al., 2024; Liu et al., 2024). In this paper, we focus on reconstructing structured indoor scenes as configurations of *planar primitives*. Among common geometric primitives such as line segments and edges, planar primitives are particularly essential for describing man-made environments. They are compact yet expressive parametric entities that provide not only strong geometric cues but also rich semantic information. Therefore, 3D maps represented by planar primitives are notably lean and convenient for practical use, and have been successfully deployed in various applications in areas such as robotics (Zhou et al., 2021; Liu et al., 2023) and augmented reality (ARKit, 2024; ARCore, 2024).

The reconstruction of 3D plane maps is conventionally approached by fitting planar primitives to an unordered point cloud or a mesh generated from depth sensors or multi-view stereo (Yu & Lafarge, 2022). The input geometries of these methods are typically noisy and incomplete, and provide only limited semantic information for robust plane detection. A recent work by Watson et al. (2024) incorporates optimized 3D semantic embeddings into the plane fitting module, *i.e.,* RANSAC (Fischler & Bolles, 1981), which helps to discern adjacent planes. However, it still heavily relies on the quality of input geometry, and requires to train from 2D plane annotations. These prior efforts basically decouple 3D plane reconstruction as two independent problems of non-plane-biased geometry recovery and plane estimation, which bottlenecks the full utilization of plane attributes. Departing from such geometry+RANSAC paradigm, Xie et al. (2022) developed the first learning-based model that incrementally reconstructs planes from posed video fragments. However, though supervised by 3D plane annotations, the method struggles with its adaptability to intricate structures.

---

*Corresponding author.

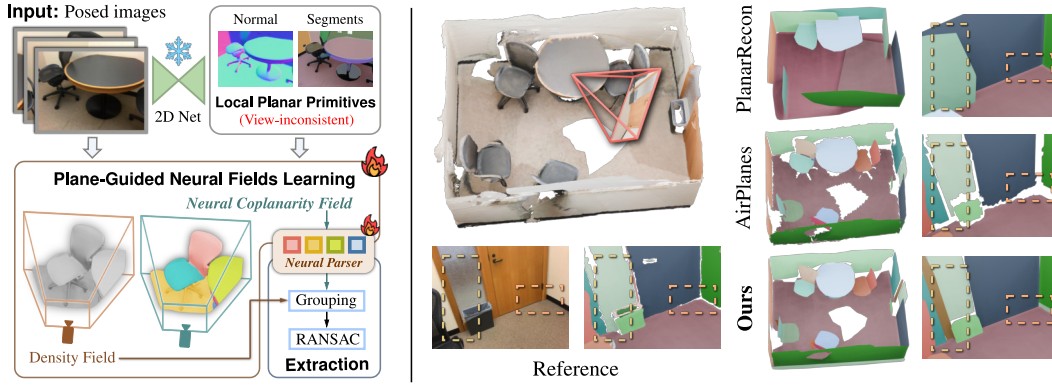

(a) Method Overview        (b) Showcasing Results of 3D Plane Reconstruction

Figure 1: **Structured 3D reconstruction in planar primitives with neural fields.**

Motivated by previous limitations, as well as the success of neural fields in sparse geometry encoding (Ye et al., 2023; Xue et al., 2024; Li et al., 2024) and 3D segmentation (Ying et al., 2024; Kim et al., 2024), we introduce a new approach named **NeuralPlane**. The proposed method maintains a consistent neural representation for environmental plane structures from multi-view inconsistent plane observations, which further enables the extraction of high-quality 3D planar primitives.

Fig. 1 *(a)* provides an overview of our pipeline, which unfolds into three phases: 1) Firstly, we combine two robust vision models into a training-free module capable of excavating well-defined plane segments from a single image. Meanwhile, plane parameters of these segments are coarsely estimated, yielding a set of locally initialized 3D entities referred to as Local Planar Primitives. 2) Secondly, we design a plane-guided training procedure where the high degree of *intra-primitive* geometric regularity is explored to optimize the density field. This can effectively mitigate the issue of poor surface reconstruction when encoding the scene geometry by volume density, however, we recognize that the density field alone is insufficient for extracting *semantically well-aligned* planar primitives. Thereby, we further propose Neural Coplanarity Field, a self-supervised feature field that involves high-level semantics. This field utilizes local planar primitives for *inter-primitive* reasoning within a contrastive learning framework, and can accurately capture the *coplanar relationships* between different regions. 3) Finally, on top of the learned neural representation, we conduct a two-stage plane extraction process: before geometrically fitting planar primitives, the neural representation is firstly grouped into isolated regions based on the learned coplanarity features, wherein a module called Neural Parser is introduced to facilitate the discrimination of plane instances.

Fig. 1 *(b)* showcases the superiority of NeuralPlane over existing state-of-the-arts (Xie et al., 2022; Watson et al., 2024), where our method faithfully rebuilds scene layouts as sets of crisp planar primitives while preserving fine-grained semantics. In summary, our key contributions are:

- We present NeuralPlane, a novel approach for multi-view 3D plane reconstruction that eschews plane annotations. It enjoys the synergy between geometry and semantics in the context of neural fields and achieves leading performance in extensive experimental studies.

- To capture accurate plane locations, a self-prompting mechanism that integrates two robust vision models is first proposed to excavate 2D plane regions, where strong plane regularity is then applied to guide the optimization of the implicitly encoded geometry.

- We enhance plane discrimination by incorporating semantics: Neural Coplanarity Field acts as the core driver of involving the semantic separation via contrastive learning, while Neural Parser is introduced to jointly model the learned coplanar relationships.

## 2 RELATED WORK

### 2.1 MULTI-VIEW PLANE RECONSTRUCTION

The basis of 3D plane reconstruction from multi-view images lies in the detection of planar regions from either raw RGB inputs or inferred dense geometries. Despite impressive progress in single

view plane detection (Yang & Zhou, 2018; Liu et al., 2019; Yu et al., 2019), a unified multi-view plane reconstruction remains challenging due to severe cross-view inconsistencies. Several attempts have been made to locally recover 3D planes from a limited number of images by predicting plane correspondences (Jin et al., 2021; Agarwala et al., 2022; Tan et al., 2023) or stereo matching (Liu et al., 2022). However, scaling to multi-view scenarios for scene-level reconstructions is still problematic. More relevant to our work, PlanarRecon (Xie et al., 2022), the first learning-based method, reconstructs 3D plane maps by incrementally detecting, tracking, and fusing 3D planes from posed video fragments. Chen et al. (2023) draw upon neural fields to detect planar primitives from RGB-D sequences. AirPlanes (Watson et al., 2024) adopts the conventional two-stage pipeline of fitting planes to non-plane-biased geometry, where the learned 2D plane priors are lifted into 3D for robust plane distinction.

## 2.2 HIGHER-LEVEL MAPPING VIA NEURAL RENDERING

Neural rendering has emerged as a powerful technique that allows distilling various 2D observations from multiple views into 3D neural fields. It has greatly reshaped a wide range of 3D scene reconstruction and understanding tasks. Recent works (Ye et al., 2023; Xue et al., 2024; Li et al., 2024) find the neural field favorable for implicitly encoding *sparse parametric primitives* such as line segments and edges. Object-compositional neural representations have also been extensively explored for *object-centric mapping* (Yang et al., 2021; Yu et al., 2022a; Wang et al., 2023; Park et al., 2024). Besides, there are advances in constructing *semantic maps* with neural fields by distilling high-dimensional features (Kobayashi et al., 2022; Kerr et al., 2023; Zhu et al., 2024) or contrastive learning (Fan et al., 2023; Bhalgat et al., 2023; Kim et al., 2024). Our study falls within the exploration of neural fields for parametric primitives. However, instead of focusing solely on geometry, we further incorporate semantics to build the neural representation at a higher level.

## 3 METHOD: NEURALPLANE

Given a set of posed images, our goal is to recover the underlying scene structure using a collection of 3D planar primitives. To this end, we first process the input images to acquire monocular plane priors known as local planar primitives (Sec. 3.1). Then, we introduce our neural scene representation, which is optimized under the plane guidance (Sec. 3.2). Once optimized, the neural representation is made explicit by our plane extraction algorithm (Sec. 3.3).

### 3.1 GENERATING LOCAL PLANAR PRIMITIVES

Essentially, a local planar primitive $P = (\mathcal{M} \subseteq I, \boldsymbol{\pi})$ is defined as a plane segment $\mathcal{M}$ of a single image $I$, associated with plane parameters $\boldsymbol{\pi} = [\boldsymbol{n}, o]$ under the world space. Here, $\boldsymbol{n} \in \mathbb{R}^3$ and $o \in \mathbb{R}^+$ are respectively the unit normal vector and the offset of the plane from the origin. Although remarkably advanced single-view plane recovery models (Tan et al., 2021; Shi et al., 2023) can be employed to estimate such local planar primitives, we will show that purely local yet robust image priors are sufficient to achieve satisfactory performance, which eliminates the need to deliberately train a model on plane annotations.

**Plane Segments from Monocular Priors.** We assume the plane segment $\mathcal{M}$ of a local planar primitive to be an image region that (1) has consistent normal directions and (2) is geometrically continuous. To excavate such plane segments, we draw inspiration from Mazur et al. (2024) to repurpose a pre-trained monocular normal predictor (Bae et al., 2021) and the powerful Segment Anything Model (SAM) (Kirillov et al., 2023). Specifically, we start by applying K-means clustering (Lloyd, 1982) to the predicted normal map, obtaining low-frequency regions with each having similar normal directions. Subsequently, SAM's automatic mask generator is employed to further segment these smooth regions by selecting the smallest mask for each prompt. This over-segmentation strategy produces semantically meaningful segments, while approximately enforcing the geometric continuity. Finally, masks exceeding a specified size threshold are chosen as the desired plane segments.

**Initializing Local Plane Geometry.** For each local planar primitive $P$, we initialize its plane normal $\boldsymbol{n}$ with $\bar{\boldsymbol{n}}$, which denotes the average of predicted normals within the plane segment. To estimate the plane offset $o$, we propose to use sparse 3D keypoints that are often available as a by-product of Structure-from-Motion (SfM) systems. Specifically, the plane offset is initialized by minimizing the

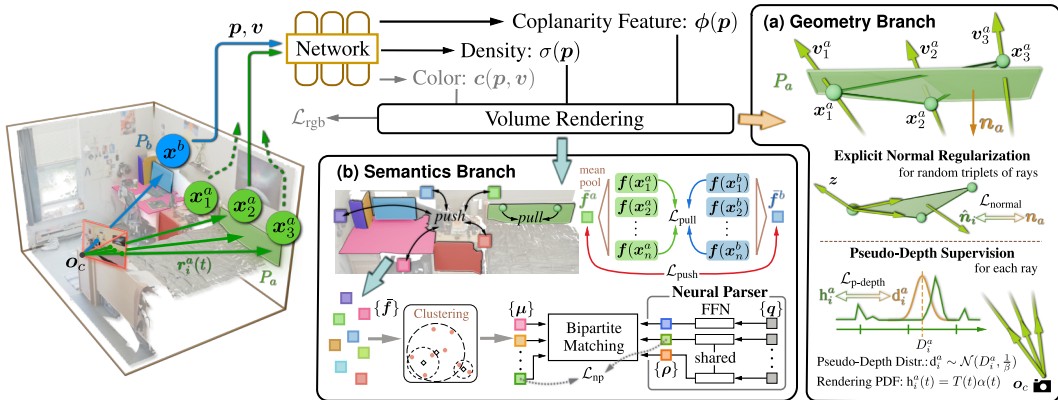

Figure 2: **Plane-guided neural fields learning.** The overall training scheme consists of two branches: *(a)* the geometry branch (Sec. 3.2.1) translates plane constraints into two intra-primitive regularization terms, *i.e.*, $\mathcal{L}_{\text{normal}}$ and $\mathcal{L}_{\text{p-depth}}$; *(b)* the semantics branch (Sec. 3.2.2) performs inter-primitive reasoning via contrastive learning, and introduces a query-based learning module (Neural Parser) to jointly model the captured coplanar relationships.

total distance to the 3D keypoints $\{\boldsymbol{p}_1, \ldots, \boldsymbol{p}_m\}$ detected in $\mathcal{M}$:

$$\bar{o} = \arg\min_{\text{v} \in \mathbb{R}^+} \sum_i^m \frac{w_i}{\sum w_i} d_i, \quad \text{where } d_i = \|\bar{\boldsymbol{n}} \cdot \boldsymbol{p}_i + \text{v}\|_2^2. \tag{1}$$

The weight $w_i = 1/\bar{e}_i$, indicating the reliability of $\boldsymbol{p}_i$, is approximated using the average reprojection error $\bar{e}_i$ across views where $\boldsymbol{p}_i$ is visible. Although error-prone, the results are sufficient to provide effective geometric guidance for our neural representation, which will be detailed below.

## 3.2 LEARNING NEURAL REPRESENTATION UNDER PLANE GUIDANCE

Note that local planar primitives are generated with minimal dependence on viewpoint, leading to severe inconsistency across views. We acknowledge that it is non-trivial to explicitly establish correspondences and merge them into a consistent 3D plane reconstruction. As a consequence, we propose to fuse them implicitly using neural fields. The overall training scheme is illustrated in Fig. 2, which consists of two major branches: 1) the geometry branch (Sec. 3.2.1) focuses on learning accurate plane locations, while 2) the semantics branch (Sec. 3.2.2) captures coplanar relationships between different regions from conflicting 2D plane segments.

### 3.2.1 GEOMETRY BRANCH

NeuralPlane implicitly encodes scene geometry as volume density. To alleviate the well-known *shape-radiance* ambiguity (Zhang et al., 2020), we translate strong plane constraints provided by prepared local planar primitives into two regularization terms.

**Preliminaries.** Standard NeRF (Mildenhall et al., 2020) represents a scene as a continuous volume density function $\sigma : \mathbb{R}^3 \mapsto [0, 1]$ and a continuous view-dependent color function $\boldsymbol{c} : (\mathbb{R}^3, \mathbb{S}^2) \mapsto [0, 1]^3$. According to the volume rendering formula, given a ray $\boldsymbol{r}$ emanating from camera center $\boldsymbol{o}_c$ with direction $\boldsymbol{v}$, its expected termination point $\boldsymbol{x}$ is computed by integrating $N$ sampled points $\{\boldsymbol{p}_i = \boldsymbol{o}_c + t_i\boldsymbol{v} | i = 1, \ldots, N, t_i < t_{i+1}\}$ along the ray:

$$\boldsymbol{x} = \sum_{i=1}^N \text{h}(t_i)\boldsymbol{p}_i, \quad \text{h}(t_i) = T(t_i)\alpha(t_i) = \left(\prod_{j=1}^{i-1}(1 - \alpha(t_j))\right)\alpha(t_i). \tag{2}$$

The piecewise-constant function $\text{h}(t_i)$ is an approximation of the rendering probability density function (PDF) to weight the contribution of each sampled point. The opacity value $\alpha(t_i)$ is computed as $\alpha(t_i) = 1 - \exp(-\sigma(\boldsymbol{p}_i)\Delta t)$.

**Explicit Normal Regularization.** To ensure normal consistency on planar surfaces, a normal regularizer $\mathcal{L}_{\text{normal}}$ is introduced to penalize the deviation of the NeRF-derived surface normal $\hat{\boldsymbol{n}}$ from $\boldsymbol{n}$ (initialized in Sec. 3.1). We estimate $\hat{\boldsymbol{n}}$ in an algebraic manner, using a triplet of rays sampled from the plane segment $\mathcal{M}$. For each ray triplet $\mathcal{T} = \{\boldsymbol{r}_i(t) = \boldsymbol{o}_c + t\boldsymbol{v}_i\}_{i=1}^3$ randomly drawn from $P_a$, we render its expected termination points $\{\boldsymbol{x}_i\}_{i=1}^3$ by eq. (2), and estimate the normal of the plane passing through:

$$\hat{\boldsymbol{n}} = -\text{sign}(\boldsymbol{z}^\top \boldsymbol{v}_1) \cdot \frac{\boldsymbol{z}}{\|\boldsymbol{z}\|}, \quad \text{where } \boldsymbol{z} = (\boldsymbol{x}_2 - \boldsymbol{x}_1) \times (\boldsymbol{x}_3 - \boldsymbol{x}_1). \tag{3}$$

Here, the direction of $\hat{\boldsymbol{n}}$ is adjusted to point towards the camera (same as $\boldsymbol{n}_a$) by checking the angle between $\boldsymbol{z}$ and $\boldsymbol{v}_1$. Then, the normal loss with respect to $P_a$ is defined as:

$$\mathcal{L}_{\text{normal}}(\sigma; P_a) = \mathbb{E}_{\mathcal{T} \sim P_a} \|1 - \hat{\boldsymbol{n}}^\top \boldsymbol{n}_a\|_1. \tag{4}$$

**Pseudo-Depth Supervision.** While the normal regularization induces smoothness in planar regions, it alone is insufficient for achieving high geometric accuracy. To further guide the density field learning, we introduce *plane-derived pseudo-depth* as extra supervision. The pseudo-depth $D$ is computed as the ray distance from the camera centre $\boldsymbol{o}_c$ to the corresponding $P$: $D = -(o + \boldsymbol{n} \cdot \boldsymbol{o}_c)/\cos\varphi$, where $\varphi$ is the angle between $\boldsymbol{n}$ and the ray direction $\boldsymbol{v}$. Since defectively estimated $\boldsymbol{n}$ and $o$ lead to noisy $D$, we follow DS-NeRF (Deng et al., 2022) to model the depth label of ray $\boldsymbol{r}$ as a random variable normally distributed around $D$ with variance $\beta^{-1}$: $\text{d} \sim \mathcal{N}(t; D, \beta^{-1})$. For each ray $\boldsymbol{r}_i$ drawn from $P_a$, we minimize the KL divergence between the pseudo-depth distribution $\mathcal{N}(t; D_i^a, \beta^{-1})$ and the piecewise-constant rendering PDF $\text{h}_i^a(t)$:

$$\mathcal{L}_{\text{p-depth}}(\sigma, \boldsymbol{\pi}_a; P_a) = \mathbb{E}_{\boldsymbol{r}_i \sim P_a} D_{\text{KL}} \left[ \mathcal{N}(t; D_i^a, \beta^{-1}) \| \text{h}_i^a(t) \right]. \tag{5}$$

In addition, we enable gradient descent over the estimated plane parameters $\boldsymbol{\pi}_a$ (*i.e.*, $\boldsymbol{n}_a$ and $o_a$), using gradient signals from $\mathcal{L}_{\text{p-depth}}$ to jointly refine the inaccurate local geometry.

### 3.2.2 SEMANTICS BRANCH

As highlighted by AirPlanes (Watson et al., 2024), semantics are crucial in 3D plane reconstruction since ideal planar primitives need to be semantically self-contained. For instance, though geometrically coplanar, a closed door should be regarded as a different planar primitive from the wall that encloses it. However, it is intractable for purely geometric methods, *e.g.,* RANSAC, to resolve such semantic conflicts. To address this, we retrofit the neural field with plane-level semantics.

**Neural Coplanarity Field.** We start by introducing Neural Coplanarity Field (NCF) $\boldsymbol{\phi} : \mathbb{R}^3 \mapsto \mathbb{R}^d$, which outputs a $d$-dimensional feature $\boldsymbol{\phi}(\boldsymbol{p})$ over any 3D location $\boldsymbol{p}$. The coplanarity between two rays is defined as the probability that their expected termination points lie on the same plane structure. Here, we aim to measure such coplanarity using the similarity between their rendered features, which we call the *coplanarity feature*: $\boldsymbol{f}(\boldsymbol{x}) = \sum_{i=1}^N \text{h}(t_i)\boldsymbol{\phi}(\boldsymbol{p}_i)$.

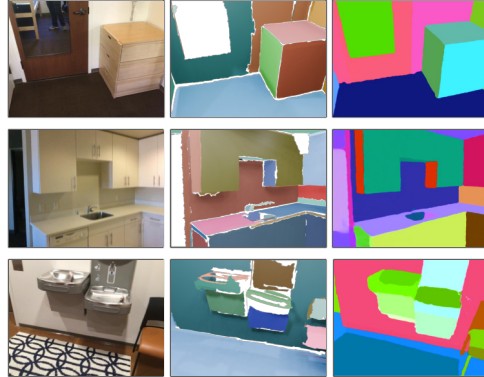

We train NCF through the margin-based contrastive learning paradigm (Chopra et al., 2005) where the objective comprises two terms: the *intra-primitive* pull loss $\mathcal{L}_{\text{pull}}$ and the *inter-primitive* push loss $\mathcal{L}_{\text{push}}$.

Considering two rays from the same local planar primitive $P_a$, *e.g.*, $\boldsymbol{r}_i$ and $\boldsymbol{r}_j$, their coplanarity features constitute a pair of positive samples and are pulled close by minimizing the $L^2$ distance:

(a) Ref. View  (b) Ground Truth  (c) Rendered

Figure 3: **Visualization of NCF**.

$$\mathcal{L}_{\text{pull}}(\boldsymbol{\phi}; P_a) = \mathbb{E}_{\{\boldsymbol{r}_i, \boldsymbol{r}_j\} \sim P_a} \|\boldsymbol{f}(\boldsymbol{x}_i) - \boldsymbol{f}(\boldsymbol{x}_j)\|_2. \tag{6}$$

Next, we define the *primitive-level* coplanarity feature of $P_a$ as: $\bar{\boldsymbol{f}}_a = \mathbb{E}_{\boldsymbol{r} \sim P_a} \boldsymbol{f}(\boldsymbol{x})$. The push loss $\mathcal{L}_{\text{push}}$ is computed over them, as we find that it can reduce computation overhead and accelerate

convergence. Note that local planar primitives are generated through over-segmentation, and $\mathcal{L}_{\text{push}}$ is computed across views, which indicates that two primitives in a batch do not necessarily belong to distinct planar structures. This ambiguity prompts us to incorporate geometry: $\bar{f}_a$ and $\bar{f}_b$ form a pair of negative samples only if the corresponding plane parameters $\pi_a$, $\pi_b$ are sufficiently far from each other. More formally, the push loss with margin $m$ is formulated as:

$$\mathcal{L}_{\text{push}}(\phi; P_a, P_b) = \mathbf{1}_{[\|o_a - o_b\| > t_o \text{ or } \|n_a \cdot n_b\| < t_n]} \cdot \text{ReLU}(m - \|\bar{f}_a - \bar{f}_b\|_2), \tag{7}$$

where $\mathbf{1}_{[\cdot]}$ is the indicator function, and $t_o$, $t_n$ are pushing thresholds w.r.t the offset and normal.

As shown in Fig. 3 (c), we visualize the rendered feature maps using PCA (Hastie et al., 2009) to demonstrate what is learned by NCF. The reference views and ground truth are also provided for comparison. The results shows that the proposed NCF can effectively capture the coplanar relationships between different regions with clear boundaries, considering both geometry and semantics.

**Neural Parser.** To obtain instance centroids and subsequently decompose the learned feature field into instances, one can apply unsupervised clustering methods to thousands of rendered features after training (Bhalgat et al., 2023). However, instead of this post-processing, we propose a learning-based module called Neural Parser to concurrently learn a set of *semantic prototypes* during training. Each prototype is assumed to be the centroid of an isotropic Gaussian distributed cluster of coplanarity features, which represents a semantically isolated subregion.

As illustrated in Fig. 2, we are motivated by NEAT (Xue et al., 2024) to adopt a query-based architecture, where $N_p$ semantic prototypes $\{\rho_i \in \mathbb{R}^d\}_{N_p}$ are predicted from $N_p$ learnable queries $\{q_i \in \mathbb{R}^{d_q}\}_{N_p}$ (randomly initialized) through a simple feed-forward network (FFN) $\theta : \mathbb{R}^{d_q} \mapsto \mathbb{R}^d$. During each training iteration, the rendered primitive-level features $\{\bar{f}\}$ are utilized to supervise the module. Specifically, we first apply DBSCAN (Ester et al., 1996) to these noisy features to obtain $N_c$ centroids $\{\mu_i \in \mathbb{R}^d\}_{N_c}$. Then, the bipartite matching is performed between the predicted prototypes $\{\rho_i\}_{N_p}$ and centroids $\{\mu_i\}_{N_c}$ using an efficient Hungarian algorithm (Jonker & Volgenant, 1987). The cost matrix is defined as the $L^2$ distance between each pair. With the optimal permutation of assignment $\tau$ determined, our objective is to keep prototypes apart while further minimizing the distance between each centroid and its matched prototype:

$$\mathcal{L}_{\text{np}}(\theta, \{q\}; \{\mu\}) = \sum_{i \neq j} \text{ReLU}(m - \|\rho_i - \rho_j\|_2) + \sum_{i=1}^{k} \|\mu_i - \rho_{\tau(i)}\|_2, \tag{8}$$

$$\text{where } k = \min(N_p, N_c), \rho_i = \theta(q_i).$$

**Overall Training Scheme.** The NeuralPlane framework is trained by assembling all loss terms:

$$\mathcal{L} = \mathcal{L}_{\text{rgb}}(\sigma) + \mathbb{E}_{P_i \sim \mathcal{P}} \left[ \lambda_1 \mathcal{L}_{\text{normal}}(\sigma; P_i) + \lambda_2 \mathcal{L}_{\text{p-depth}}(\sigma, \pi_i; P_i) + \lambda_3 \mathcal{L}_{\text{pull}}(\phi; P_i) \right]$$
$$+ \mathbb{E}_{\{P_i, P_j\} \sim \mathcal{P}} \left[ \mathcal{L}_{\text{push}}(\phi; P_i, P_j) \right] + \mathcal{L}_{\text{np}}(\theta, \{q\}; \{\mu\}), \tag{9}$$

where $\mathcal{L}_{\text{rgb}}$ is the standard NeRF photometric loss, and the set $\mathcal{P}$ denotes all local planar primitives. We set the balancing parameters $\lambda_1$, $\lambda_2$, $\lambda_3$ to 0.01, 0.1 and 0.5, respectively.

## 3.3 Extracting Global 3D Parametric Planes

Once the implicit neural representation is optimized to encode both the geometry and semantics of the scene, a straightforward 3D plane extraction method is employed to make the representation explicit. The process involves five steps: (1) render sufficient termination points and their coplanarity features from training views; (2) assign the label of the nearest semantic prototype to each point based on its rendered coplanarity feature; (3) group these points according to their labels; (4) parametrically fit 3D planes to each point group using RANSAC; (5) project the inliers onto each fitted plane and triangulate the rasterized projection into a mesh, which represents the final planar primitive. We refer the reader to Appendix A.1 for more details.

## 4 Experiments

### 4.1 Experimental Setup

**Datasets.** To evaluate NeuralPlane's ability to reconstruct 3D plane maps, we conduct experiments on 12 challenging real-world indoor scenes: 8 scenes from ScanNetv2 (Dai et al., 2017) and 4 scenes

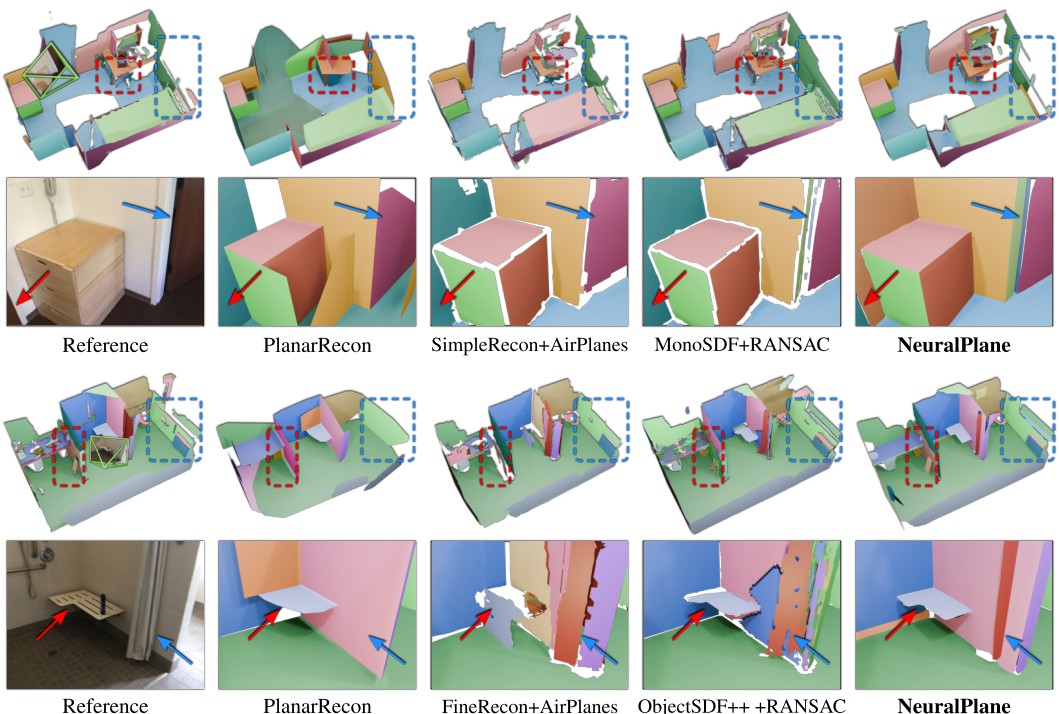

Figure 4: **Qualitative results on ScanNetv2.** For each scene, the first row presents the top view of the entire room, and the second row displays the details viewed from a given pose. Only the best two geometry+RANSAC methods are included for comparison. NeuralPlane reconstructs clean planar structures with fine details and coherent semantics.

from ScanNet++ (Yeshwanth et al., 2023). Following standard practice (Xie et al., 2022), we adopt scripts provided by PlaneRCNN (Liu et al., 2019) to generate ground-truth 3D planes for evaluation.

**Baselines.** Our NeuralPlane is evaluated against the learning-based multi-view plane reconstruction approach PlanarRecon (Xie et al., 2022), as well as a wide variety of geometry+RANSAC methods.

The geometry of the latter is inferred either by learning-based Multi-View Stereo (MVS) methods, including SimpleRecon (Sayed et al., 2022) and FineRecon (Stier et al., 2023), or by representative indoor neural surface reconstruction methods (Wang et al., 2021) listed as follows: (1) ManhattanSDF (Guo et al., 2022), which improves the reconstruction quality by enforcing the Manhattan-world assumption to planar regions like walls and floors; (2) NeuRIS (Wang et al., 2022), which utilizes predicted normal priors to adaptively guide the optimization of neural representations; (3) MonoSDF (Yu et al., 2022b), which uses both learned monocular depth and normal priors; (4) ObjectSDF++ (Wu et al., 2023a), which extends MonoSDF by incorporating dense instance annotations for high-fidelity object-compositional surface reconstruction. We also include the traditional surface reconstruction method (Schönberger et al., 2016; Kazhdan & Hoppe, 2013) implemented in COLMAP (Schönberger & Frahm, 2016). Given meshes produced by these methods, we run Sequential RANSAC or its variant, namely AirPlanes (Watson et al., 2024), to partition them into planar primitives. For all data-driven components, we use the official pre-trained models.

**Metrics.** We evaluate the geometric quality of 3D plane reconstruction utilizing standard 3D metrics introduced by Bozic et al. (2021). Among these metrics, we adopt *Chamfer Distance* and *F-score* with a threshold of $5\,\mathrm{cm}$ as the comprehensive metrics for comparison. To assess how well the reconstructed planar primitives align with the ground-truth scene semantics, we further employ three conventional segmentation metrics (Arbeláez et al., 2011; Tan et al., 2021; Shi et al., 2023): rand index (RI), variation of information (VOI) and segmentation covering (SC).

**Implementation Details.** NeuralPlane is implemented in Nerfstudio (Tancik et al., 2023) on top of Nerfacto, a unified approach in the literature of NeRF. Considering the lack of texture in typical

Table 1: **Quantitative results on ScanNetv2.** Top-3 results are highlighted as first, second and third. At the top, we also report the quality of raw geometries in 7 baseline methods, using official scene reconstructions as the ground truth. † is tested with its MLP variant. ‡ produces a set of semantically decomposed reconstructions, and Seq.RANSAC is executed on each of them independently.

| Method | | Geometry | | | | | | Segmentation | | |
|---|---|---|---|---|---|---|---|---|---|---|
| | | Accu. ↓ | Comp. ↓ | **Chamfer** ↓ | Prec. ↑ | Recall ↑ | **F-score** ↑ | RI ↑ | VOI ↓ | SC ↑ |
| COLMAP (Schönberger et al., 2016) | | 18.35 | 13.40 | **15.88** | 40.5 | 42.4 | **41.2** | - | - | - |
| SimpleRecon (Sayed et al., 2022) | | 6.74 | 5.55 | **6.15** | 59.2 | 58.8 | **59.0** | - | - | - |
| FineRecon (Stier et al., 2023) | | 6.12 | 4.19 | **5.16** | 68.6 | 72.9 | **70.6** | - | - | - |
| ManhattanSDF (Guo et al., 2022) | | 9.95 | 7.51 | **8.73** | 50.4 | 50.9 | **50.6** | - | - | - |
| NeuRIS (Wang et al., 2022) | | 10.23 | 5.69 | **7.96** | 62.0 | 64.8 | **63.2** | - | - | - |
| MonoSDF† (Yu et al., 2022b) | | 5.28 | 5.07 | **5.18** | 69.7 | 69.7 | **69.7** | - | - | - |
| ObjectSDF++† (Wu et al., 2023a) | | 8.82 | 5.25 | **7.03** | 54.9 | 68.9 | **60.9** | - | - | - |
| COLMAP | | 19.32 | 13.37 | **16.34** | 40.6 | 40.9 | **40.6** | 0.928 | 3.91 | 0.154 |
| SimpleRecon | | 6.24 | 6.00 | **6.12** | 57.7 | 52.6 | **54.9** | 0.949 | 2.65 | 0.272 |
| FineRecon | | 5.20 | 5.65 | **5.43** | 69.1 | 64.9 | **66.7** | 0.941 | 2.56 | 0.276 |
| ManhattanSDF | +Seq.RANSAC | 9.37 | 8.70 | **9.04** | 50.6 | 51.0 | **50.8** | 0.930 | 2.82 | 0.281 |
| NeuRIS | | 9.87 | 6.35 | **8.11** | 59.6 | 59.3 | **59.3** | 0.945 | 2.57 | 0.293 |
| MonoSDF | | 5.91 | 5.43 | **5.67** | 65.9 | 66.1 | **65.9** | 0.945 | 2.38 | 0.333 |
| ObjectSDF++‡ | | 8.41 | 5.35 | **6.88** | 58.0 | 68.8 | **62.8** | 0.952 | 2.32 | 0.334 |
| SimpleRecon | | 5.43 | 6.60 | **6.01** | 59.2 | 51.5 | **55.1** | 0.944 | 2.51 | 0.341 |
| FineRecon | +AirPlanes (Wat- | 4.93 | 5.95 | **5.44** | 70.8 | 62.2 | **66.2** | 0.947 | 2.43 | 0.310 |
| ManhattanSDF | son et al., 2024) | 9.46 | 9.30 | **9.38** | 52.7 | 50.2 | **51.3** | 0.940 | 2.61 | 0.315 |
| NeuRIS | | 6.05 | 7.38 | **6.71** | 66.2 | 56.7 | **61.0** | 0.943 | 2.55 | 0.291 |
| MonoSDF | | 4.57 | 6.33 | **5.45** | 72.2 | 62.0 | **66.6** | 0.948 | 2.38 | 0.346 |
| PlanarRecon (Xie et al., 2022) | | 7.74 | 11.85 | **9.80** | 55.2 | 44.3 | **49.0** | 0.909 | 3.27 | 0.265 |
| **NeuralPlane@PlaneRecTR** | | 5.38 | 4.65 | **5.02** | 67.6 | 70.0 | **68.7** | 0.949 | 2.37 | 0.364 |
| **NeuralPlane** | | 4.92 | 4.27 | **4.59** | 70.5 | 71.9 | **71.2** | 0.955 | 2.25 | 0.376 |

indoor scenes, we use the COLMAP toolbox (Sarlin et al., 2019) that supports LoFTR (Sun et al., 2021), to export dense 3D keypoints for the initialization of local planar primitives.

We train NeuralPlane for 4k iterations with batches of 8192 rays across all scenes. For the first 1k iterations, we nullify $\mathcal{L}_{\text{p-depth}}$ and $\mathcal{L}_{\text{push}}$ on local planar primitives that observe fewer than 50 3D keypoints. After the density field is holistically optimized to a decent state, we re-estimate the plane offsets of these nullified primitives using rendered depth, globally activate the overall loss in eq. (9), and enable the local geometry refinement. We list several key hyperparameters used in our main experiments: the dimension of coplanarity feature $d = 4$, the pushing thresholds $(t_o, t_n) = (8\,\text{cm}, \cos 10°)$, and the number of semantic prototypes $N_p = 32$. A fixed set of hyperparameters is employed across all test scenes. Preprocessing local planar primitives takes around 2 to 5 minutes, followed by about 6 minutes for training on a single NVIDIA RTX 3090 GPU.

## 4.2 EVALUATION

**ScanNetv2.** We select 8 challenging scenes with diverse layouts to benchmark the adaptability of various methods. To demonstrate the superiority of our local planar primitives generation method introduced in Sec. 3.1, we include a variant denoted by *NeuralPlane@PlaneRecTR*, where the initial local planar primitives are instead produced by PlaneRecTR (Shi et al., 2023). As reported in Tab. 1, NeuralPlane excels existing methods across all metrics. For geometry+RANSAC methods, we observe that AirPlanes (Watson et al., 2024) in combination with FineRecon (Stier et al., 2023) or MonoSDF (Yu et al., 2022b) exhibits impressive results in Precision but suffers a significant drop in Recall. This indicates that incorporating learned plane embeddings can effectively avoid the undesirable fusion of adjacent planes and thus better fit the input geometry (high precision). However, the involvement of plane embeddings during RANSAC also incurs the risk of missing planes, resulting in more false negatives (low recall). In contrast, NeuralPlane strikes a good balance between Precision and Recall. In addition, we observe that independently applying RANSAC to each decomposed mesh of ObjectSDF++, which is optimized with *ground-truth instance annotations*, yields improved geometry and segmentation. However, NeuralPlane achieves even better segmentation with only *machine-predicted* masks. See Fig. 4 for qualitative evaluation.

Table 2: **Quantitative results on ScanNet++.** Top-3 results are highlighted as first, second and third. † is tested using its Multi-Res Grids variant.

| Method | | Geometry | | | | | | Segmentation | | |
|---|---|---|---|---|---|---|---|---|---|---|
| | | Accu. ↓ | Comp. ↓ | **Chamfer** ↓ | Prec. ↑ | Recall ↑ | **F-score** ↑ | RI ↑ | VOI ↓ | SC ↑ |
| COLMAP | | 24.42 | 14.53 | **19.47** | 47.4 | 44.2 | **45.4** | 0.920 | 3.91 | 0.160 |
| SimpleRecon | | 11.03 | 8.54 | **9.79** | 43.9 | 45.8 | **44.6** | 0.936 | 3.17 | 0.195 |
| FineRecon | +Seq.RANSAC | 4.70 | 6.03 | **5.36** | 80.3 | 71.1 | **75.3** | 0.929 | 2.79 | 0.252 |
| ManhattanSDF | | 8.72 | 8.67 | **8.70** | 59.4 | 56.3 | **57.7** | 0.928 | 3.06 | 0.248 |
| NeuRIS | | 5.35 | 4.33 | **4.84** | 81.4 | 80.4 | **80.9** | 0.941 | 2.46 | 0.315 |
| MonoSDF† | | 5.59 | 4.60 | **5.09** | 77.0 | 78.5 | **77.7** | 0.939 | 2.47 | 0.288 |
| SimpleRecon | | 7.97 | 9.84 | **8.91** | 46.5 | 42.1 | **44.1** | 0.931 | 2.90 | 0.219 |
| FineRecon | | 3.21 | 7.54 | **5.37** | 84.0 | 68.7 | **75.5** | 0.941 | 2.66 | 0.277 |
| ManhattanSDF | +AirPlanes | 7.35 | 9.84 | **8.59** | 57.9 | 52.0 | **54.5** | 0.935 | 2.85 | 0.267 |
| NeuRIS | | 2.99 | 6.38 | **4.69** | 87.3 | 73.8 | **79.9** | 0.943 | 2.53 | 0.287 |
| MonoSDF† | | 3.21 | 7.38 | **5.29** | 82.8 | 67.6 | **74.2** | 0.935 | 2.69 | 0.264 |
| PlanarRecon | | 7.91 | 20.67 | **14.29** | 53.1 | 38.0 | **43.8** | 0.900 | 3.49 | 0.231 |
| **NeuralPlane@PlaneRecTR** | | 6.03 | 6.31 | **6.17** | 71.1 | 69.1 | **70.0** | 0.939 | 2.72 | 0.301 |
| **NeuralPlane** | | 4.33 | 4.87 | **4.60** | 80.8 | 78.7 | **79.7** | 0.950 | 2.38 | 0.356 |

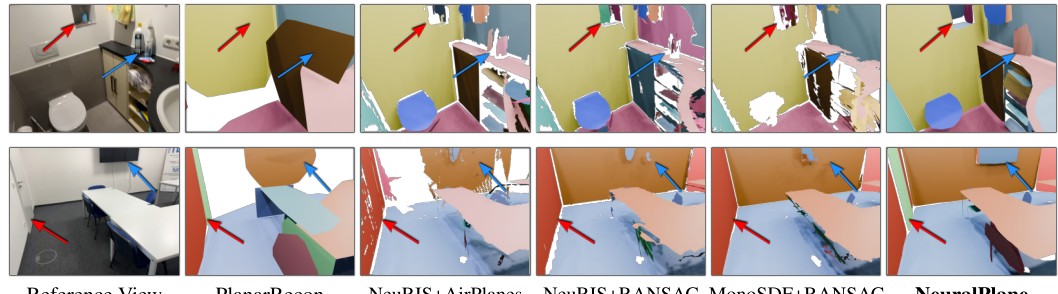

| Reference View | PlanarRecon | NeuRIS+AirPlanes | NeuRIS+RANSAC | MonoSDF+RANSAC | **NeuralPlane** |

Figure 5: **3D plane reconstruction results on ScanNet++.**

**ScanNet++.** We further conduct experiments on 4 scenes from ScanNet++ (Yeshwanth et al., 2023), noting that no pre-trained model adopted in our method or baselines has encountered similar scene layouts before. These scenes are challenging due to the significant absence of textures. We observe that low-texture regions generate normal priors that provide NeuRIS with relatively precise geometric guidance, but result in sparse and misleading SfM keypoints for our method. Despite this, as reported in Tab. 2, our approach still achieves competitive geometric performance. Moreover, our method consistently demonstrates remarkable segmentation performance, preserving fine-grained and coherent semantics (see qualitative results in Fig. 5). It is also worth noting that in our method, geometry is directly represented by volume density, which allows for a 40× training speedup compared to other neural implicit methods. A more detailed comparison of reconstruction efficiency is provided in Appendix A.4.

Table 3: **Ablation study on proposed components.** NCF and NP denote the proposed neural coplanarity field and neural parser module, respectively. The best results are in **bold**.

| Method | Components | | | | | Geometry | | Segmentation | | |
|---|---|---|---|---|---|---|---|---|---|---|
| | $\mathcal{L}_{normal}$ | $\mathcal{L}_{p\text{-}depth}$ | Refine. | NCF | NP | Chamfer ↓ | F-score ↑ | RI ↑ | VOI ↓ | SC ↑ |
| Nerfacto | - | - | - | - | - | 19.61 | 17.3 | 0.903 | 4.66 | 0.119 |
| Model A | - | ✓ | ✓ | ✓ | ✓ | 4.96 | 68.1 | 0.951 | 2.27 | 0.371 |
| Model B | ✓ | - | ✓ | ✓ | ✓ | 10.01 | 41.9 | 0.935 | 2.98 | 0.305 |
| Model C | ✓ | ✓ | - | ✓ | ✓ | 5.02 | 66.0 | 0.944 | 2.44 | 0.358 |
| Model D | ✓ | ✓ | ✓ | - | - | 4.98 | 66.1 | 0.940 | 2.59 | 0.352 |
| Model E | ✓ | ✓ | ✓ | ✓ | - | 4.91 | 69.8 | 0.947 | 2.40 | 0.364 |
| NeuralPlane | ✓ | ✓ | ✓ | ✓ | ✓ | **4.59** | **71.2** | **0.955** | **2.25** | **0.376** |

### 4.3 ABLATION STUDY

**Ablating Components.** To verify the merits of each component in NeuralPlane, we conduct ablations using the 8 scenes from ScanNetv2. Quantitative results are listed in Tab. 3 , where Nerfacto is our backbone with default settings and in Model E, the neural parser is replaced by post-clustering similar to ContrastiveLift (Bhalgat et al., 2023). Nerfacto and models A through C all struggle with degraded reconstruction quality due to insufficient geometric guidance. Model D disregards high-level semantics during the plane extraction, resulting in significant semantic conflicts as depicted in Fig. 6 (d). NeuralPlane excels all these models on all metrics, demonstrating that geometry and semantics are tightly entangled in the task, and that the combination in our method appears notable synergy.

**Ablating Number of Semantic Prototypes.** Remember that the number of semantic prototypes $N_p$ is a heuristically determined hyperparameter in neural parser and we opt for $N_p = 32$ across all scenes in our main experiments. We assess the effect of varying $N_p$ on the reconstruction results and observe that both geometry and segmentation deteriorate as $N_p$ continues to increase (see Fig. 7 (a)). This could be explained by the inherent ambiguity in defining what constitutes a single plane structure: in Fig. 7 (b), we illustrate that by controlling $N_p$, one can adjust the level of detail (LOD), while the level with $N_p = 32$ is potentially closer to the ground truth used for evaluation.

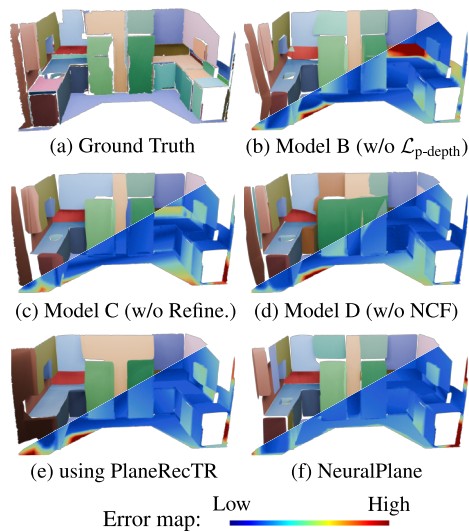

(a) Ground Truth    (b) Model B (w/o $\mathcal{L}_{\text{p-depth}}$)

(c) Model C (w/o Refine.)    (d) Model D (w/o NCF)

(e) using PlaneRecTR    (f) NeuralPlane

Error map: Low ▬▬▬ High

Figure 6: **Qualitative ablation on components.** The upper part shows the reconstructed 3D planes in different colors. The lower part presents the error map.

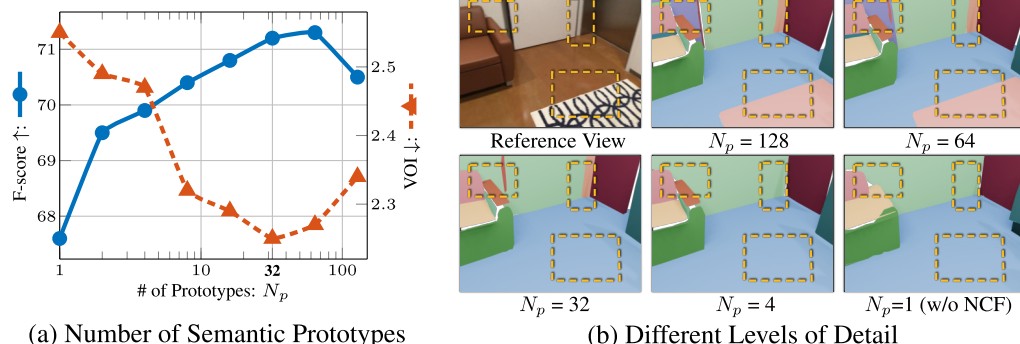

(a) Number of Semantic Prototypes

Reference View    $N_p = 128$    $N_p = 64$

$N_p = 32$    $N_p = 4$    $N_p = 1$ (w/o NCF)

(b) Different Levels of Detail

Figure 7: **Ablation study on number of semantic prototypes.** *(a)* There is a minor decline in performance as $N_p$ grows. *(b)* A case shows that the level of detail is controlled by $N_p$.

### 5 CONCLUSION

Reconstructing man-made environments as arrangements of planar primitives demands not only geometric fidelity but also semantic alignment. In this paper, we introduced NeuralPlane, a novel plane reconstruction pipeline that learns accurate 3D locations and instance-level semantics of plane structures from multi-view plane observations implicitly using neural fields. We presented an annotation-free module that incorporates strong single image priors to identify well-defined 2D plane segments, and demonstrated how these segments can drive the joint modeling of scene geometry and semantics. Our method achieves remarkable performance in delivering high-quality planar primitives with fine details and coherent semantics. Besides, our approach shows the potential in adjusting the level of detail, which we believe is practical and will provide inspiration for future research.

ACKNOWLEDGMENTS

We would like to thank Yuxin Cao and Qiran Qian for their feedback on drafts. This work was supported in part by the Beijing Natural Science Foundation (No. L223003), the National Natural Science Foundation of China (No. U22B2055, 62273345 and 62402495), and the Key R&D Project in Henan Province (No. 231111210300).

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

# A APPENDIX

In Appendix A.1, we provide additional details about the method, including the procedures of generating 2D plane segments and extracting 3D parametric planes. Next, we present further implementation details, including the network architecture and training configurations (Appendix A.2). Then, detailed descriptions on the data processing steps, implementation details of baseline methods and the evaluation protocol are provided in Appendix A.3. More experimental evaluation and qualitative results can be found in Appendix A.4. Finally, in Appendix A.5, we discuss the limitations of our method and anticipate future research trajectories.

## A.1 ADDITIONAL METHOD DETAILS

**2D Plane Segments from Monocular Priors.** In Sec. 3.1, when identifying plane regions from a single image, we first apply K-means clustering (K=8) to its estimated normal map, and select the largest 6 clusters as initial masks (see Fig. 8 *(c)* ), which correspond to low-frequency regions. We then utilize the zero-shot capacities of the Segment Anything Model (SAM) to generate all possible plane instances by prompting with 256 query points evenly sampled from the initial masks. SAM-generated masks are filtered using post-processing techniques including stability checks and non-maximal suppression (NMS) introduced by Kirillov et al. (2023). After eliminating unstable and duplicate masks, we only keep the smallest mask for each query. The overlaps between the SAM-generated masks and the initial masks, expected to be geometrically continuous and smooth, represent the desired 2D plane segments (see Fig. 8 *(d)* ). We further remove noisy 2D plane segments whose pixel areas are less than 0.4 % of the image size.

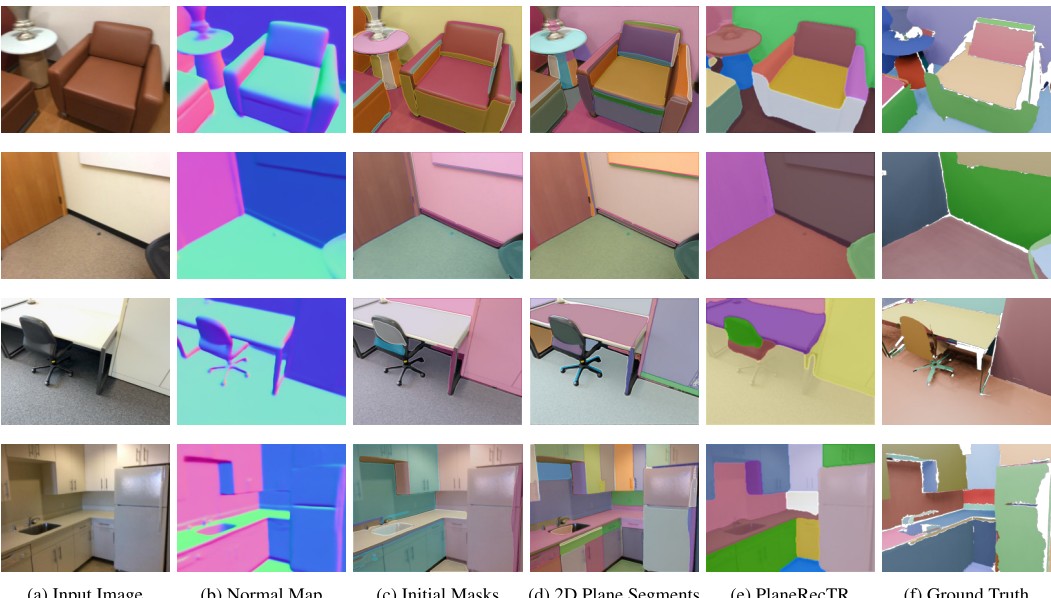

| (a) Input Image | (b) Normal Map | (c) Initial Masks | (d) 2D Plane Segments | (e) PlaneRecTR | (f) Ground Truth |

Figure 8: **2D plane segment generation.** We cluster *(b)* monocular normal priors for *(c)* initial masks, and then enforce geometric continuity by semantic over-segmentation for *(d)* 2D plane segments. *(e)* Single-view plane segmentation results by PlaneRecTR (Shi et al., 2023). *(f)* Ground truth rendered from 3D plane annotations.

Compared to the PlaneRecTR's predictions in Fig. 8 *(e)* and the ground truth in Fig. 8 *(f)*, our method exhibits over-segmentation; in other words, it tends to split a single plane into multiple segments due to SAM's fine-grained visual understanding capability.

**Pseudo-Depth Supervision.** In Sec. 3.2.1, *plane-derived pseudo-depth* is introduced as extra geometric guidance. The expected pseudo-depth $D$ of a ray $r$ from local planar primitive $P$ is computed base on the plane parameters $\boldsymbol{\pi} = [\boldsymbol{n}, o]$ (defined in world space). More precisely, let the expected termination point of $r$ be denoted by $\boldsymbol{x} = \boldsymbol{o}_c + D\boldsymbol{v}$, where $\boldsymbol{o}_c$ is the camera center and $\boldsymbol{v}$ is the unit

direction vector. Since $r$ is assumed to terminate at the planar surface $\pi$, we have

$$n \cdot (o_c + Dv) + o = 0, \tag{10}$$

which can be rewritten as

$$D = \frac{-(o + n \cdot o_c)}{\langle n, v \rangle} = \frac{-(o + n \cdot o_c)}{\cos \varphi}, \tag{11}$$

where $\varphi$ is the angle between $n$ and $v$.

Motivated by DS-NeRF (Deng et al., 2022), the pseudo-depth loss defined in eq. (5) encourages the termination distribution of a ray to match the pseudo-depth distribution. Different from the paper, we allow gradients to flow through the pseudo-depth distribution, thereby jointly optimizing $n$ and $o$. Here, the KL divergence is computed as:

$$D_{\mathrm{KL}}\left[\mathcal{N}(t; D, \beta^{-1}) \| \mathrm{h}(t)\right] = \int \mathcal{N}(t; D, \beta^{-1}) \log \left[\frac{\mathcal{N}(t; D, \beta^{-1})}{\mathrm{h}(t)}\right] dt \tag{12}$$

$$\approx \sum_j \mathcal{N}(t_j; D, \beta^{-1}) \log \left[\frac{\mathcal{N}(t_j; D, \beta^{-1})}{\mathrm{h}(t_j)}\right] \Delta t_j, \tag{13}$$

where $\mathcal{N}(t; D, \beta^{-1})$ denotes the normal distribution that are used to model the noisy pseudo-depth variable d, and $\mathrm{h}(t)$ is the piecewise-constant approximation of the rendering probability density function defined in eq. (2). Then, we have:

$$D_{\mathrm{KL}}[\mathcal{N}(t; D, \beta^{-1}) \| \mathrm{h}(t)]$$
$$\propto \sum_j \exp\left(\frac{-\beta(t_j - D)^2}{2}\right) \left(\log \sqrt{\frac{\beta}{2\pi}} - \frac{\beta(t_j - D)^2}{2} - \log \mathrm{h}(t_j)\right) \Delta t_j. \tag{14}$$

The variance parameter $\beta^{-1}$ is fixed to 0.005 across all rays.

**3D Parametric Plane Extraction.** After training, in Sec. 3.3, we design a straightforward algorithm to extract 3D parametric planes from the learned neural representation. More specifically, for a given local planar primitive $P$, we randomly sample 128 rays from its 2D plane segment and render their termination points. The normal of $P$ is also assigned to each termination point as its local normal. To render the coplanarity features more efficiently, instead of sampling and integrating features along the rays, we simply use features queried at the termination locations as the ray-level coplanarity features. To group these termination points into different subregions, we first find the closest semantic prototype to each coplanarity feature by measuring the Euclidean distance in feature space. If more than half of the features share the same nearest semantic prototype, then we assign the label of that prototype to all the points. Otherwise, this local planar primitive is considered ambiguous and discarded as an outlier. We continue sampling until the number of successfully labeled points exceeds $2 \times 10^6$.

After grouping these points by their labels, we sequentially fit 3D planes to each group using RANSAC. A point is considered an inlier if: (1) the angle between its normal and the normal hypothesis is less than 20°, and (2) the distance from the point to the plane is less than 0.08 m.

Next, given a collection of fitted plane parameters and inliers, we transform them into triangulated planar surfaces by:

(1) Downsample the inliers using voxel grid filtering with a leaf size of 1 cm.

(2) Project the downsampled points onto the fitted plane and rasterize the 2D projection map into a regular binary grid with a resolution of 2 cm.

(3) Traverse the binary grid and triangulate by connecting occupied neighbors, which produces a 2D mesh.

(4) Backproject the 2D mesh into the original 3D space to obtain a 3D mesh which represents the final real-world planar structure.

## A.2 ADDITIONAL IMPLEMENTATION DETAILS

**Network Architecture.** The neural coplanarity field is configured as a separate output head along-side the default Nerfacto model (see Fig. 9). The coplanarity features are encoded by a 12-level hashgrid with resolutions sampled from 16 to 256, and a 4-layer MLP with hidden dimensions of 256 and ReLU activation. To speed up the training process, we follow GARField (Kim et al., 2024) to first render the hash value and then feed it into the coplanarity feature MLP. In addition, we constrain coplanarity features to a unit hypersphere. The feed-forward network $\theta$ employed in Neural Parser is a 3-layer MLP with both input and hidden dimensions of 8 and ReLU activation.

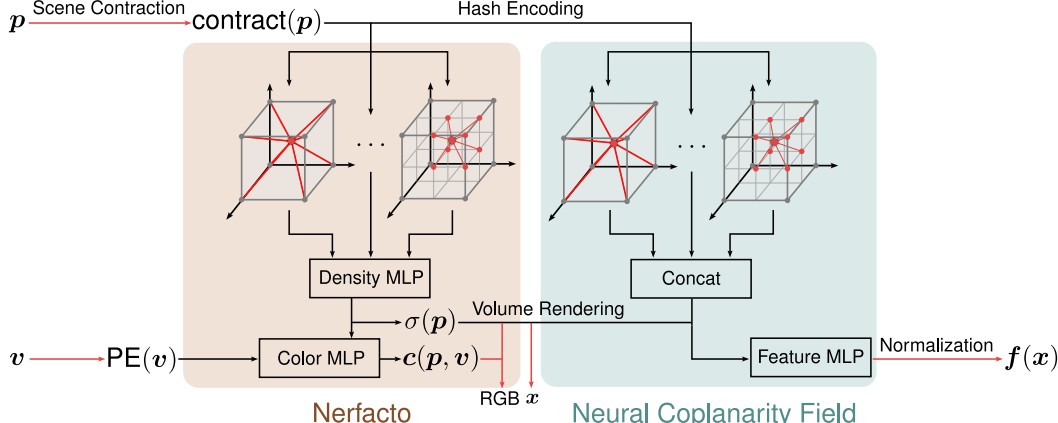

Figure 9: **Network architecture.**

**Training Details.** Before training, camera poses are centralized without scale adjustment, and we disable joint pose optimization using the provided poses. During each training iteration, we randomly select 128 local planar primitives with probabilities weighted by the sizes of corresponding 2D plane segments. In each local planar primitive, we uniformly sample $n = 8192 / 128 = 64$ rays, with $N=48$ points sampled per ray. We further randomly generate 64 triplets and 64 pairs of rays to obtain explicit surface normals for $\mathcal{L}_{\text{normal}}$ defined in eq. (4) and positive samples for $\mathcal{L}_{\text{pull}}$ defined in eq. (6), respectively. The push margin $m$ in eq. (7) is set to 1.5 for the first 1k iterations, and is later increased to 2. In Neural Parser, the DBSCAN epsilon is fixed to 0.2. We optimize plane parameters of local planar primitives using Adam optimizer with an exponential decay schedule from an initial learning rate of $1 \times 10^{-3}$ to $1 \times 10^{-4}$.

## A.3 MORE DETAILS ON EXPERIMENTS

### A.3.1 DATASETS

In our experiments, two publicly available datasets, ScanNetv2 (Dai et al., 2017) and ScanNet++ (Yeshwanth et al., 2023), are used to evaluate our method as they offer surface reconstructions and densely annotated instance-level semantics that can be further processed (Liu et al., 2019) to obtain the ground truth. Specifically, we sample 8 scenes from the official validation set of ScanNetv2 and 4 scenes from ScanNet++. All frames are resized to a resolution of $640 \times 480$. For each scene from ScanNetv2, one-eighth of the frames are uniformly selected for reconstruction, while in ScanNet++, equally spaced frames are downsampled according to the number of registered frames, leaving 150~279 frames per scene. Tab. 4 lists more details of the selected 12 scenes.

### A.3.2 IMPLEMENTATION DETAILS ON BASELINE METHODS

All the baseline methods are implemented following the official codebases and instructions. FineRecon (Stier et al., 2023) recovers accurate surfaces by introducing a depth guidance strategy, and here we use the multi-view depth predicted by SimpleRecon (Sayed et al., 2022) for such additional prior. Besides, as pointed by Yu et al. (2022b), the MLP architecture is inherently robust to motion blur and noisy camera poses due to its "smoothness bias". Thus, we adopt the MLP variant of MonoSDF

Table 4: **Details of selected scenes.** For each scene, we downsample the video to a lower frame rate. *Down. Rate* denotes the downsample rate. *Area* is the surface area of ground-truth mesh.

| | Scene ID | Type | # of Frames | Down. Rate | Area ($m^2$) |
|---|---|---|---|---|---|
| ScanNetv2 | 0084_00 | bathroom | 247 | 12.5 % | 42.68 |
| | 0164_03 | kitchen | 140 | 12.5 % | 28.97 |
| | 0217_00 | bedroom | 156 | 12.5 % | 40.65 |
| | 0316_00 | lounge | 97 | 12.5 % | 26.56 |
| | 0356_00 | bedroom | 170 | 12.5 % | 27.77 |
| | 0427_00 | conference room | 165 | 12.5 % | 24.11 |
| | 0488_01 | kitchen | 137 | 12.5 % | 38.53 |
| | 0568_00 | lounge | 207 | 12.5 % | 71.49 |
| ScanNet++ | f6659a3107 | conference room | 181 | 1.8 % | 133.38 |
| | 31a2c91c43 | bathroom | 184 | 2.4 % | 31.72 |
| | 7bc286c1b6 | bathroom | 150 | 3.3 % | 25.13 |
| | 303745abc7 | office | 279 | 4.4 % | 63.43 |

for experiments on ScanNetv2, while in ScanNet++ with high-quality RGB captures, we use the Multi-Res.Grid variant for faster convergence.

The Sequential RANSAC employed in baselines and our method is all adopted from Planar-Recon (Xie et al., 2022), with thresholds specifically fine-tuned for each method. We render per-pixel depth map for each RGB keyframe using geometries given by various baseline surface reconstruction methods so as to combine them with AirPlanes (Watson et al., 2024).

### A.3.3 EVALUATION PROTOCOL

As several methods such as FineRecon (Yu & Lafarge, 2022) can predict geometry for unseen regions, we mask out these regions via a visibility check, only preserving the observed areas for a fair evaluation.

Tab. 5 *(a)* lists the metrics in geometry, where $\mathbb{P}$ and $\mathbb{P}^*$ are point clouds evenly sampled from the predicted and ground-truth mesh, each consisting of 200 000 points. To assess how well the plane reconstruction aligns with the ground-truth semantic information, we follow the evaluation protocol of PlanarRecon (Xie et al., 2022), reporting segmentation quality using three typical metrics. These metrics are listed in Tab. 5 *(b)* and were originally used for comparing partitions (Meila, 2005; Arbeláez et al., 2011). The ground-truth clustering $\mathbb{C}^*$ is the partition of the ground-truth mesh into sets $\mathcal{C}_1^*, \mathcal{C}_2^*, \cdots, \mathcal{C}_{K^*}^*$, where $C_i^*$ represents the collection of vertices of the $i$-th ground-truth planar primitive. The predicted clustering $\mathbb{C}$ is obtained by partitioning *ground-truth vertices* according to the plane IDs of their nearest neighbors in the *predicted mesh*. The total number of vertices in the ground-truth mesh is denoted by $N_{\mathbb{C}}$.

- *Rand Index* represents the probability that $\mathbb{C}$ and $\mathbb{C}^*$ agree on the clustering of a randomly selected pair of vertices, where $N_{\text{disagree}}$ is defined as the number of pairs that are clustered differently by $\mathbb{C}$ and $\mathbb{C}^*$.

- *Variation of Information*, the sum of the entropies of $\mathbb{C}$ and $\mathbb{C}^*$ minus the mutual information between them, measures the distance between two partitions by information difference.

- *Segmentation Covering* is the average of mutual *coverings* between $\mathbb{C}$ and $\mathbb{C}^*$, where the covering $C(\mathbb{C}^* \to \mathbb{C})$ is computed as:

$$C(\mathbb{C}^* \to \mathbb{C}) = \frac{1}{N_{\mathbb{C}}} \sum_{\mathcal{C} \in \mathbb{C}} |\mathcal{C}| \cdot \max_{\mathcal{C}^* \in \mathbb{C}^*} \text{IoU}(\mathcal{C}, \mathcal{C}^*).$$

### A.4 MORE EXPERIMENTAL RESULTS

**Ablating Feature Dimension of NCF.** We investigate the performance of our proposed NeuralPlane when using different feature dimensions $d$. Specifically, we evaluate the effect of varying $d$ from

Table 5: **Definitions of metrics.** The quality of 3D plane reconstruction is evaluated by both geometry and segmentation metrics.

(a) Evaluation Metrics for Geometry.

| Metric | Definition |
|---|---|
| Accuracy | $\text{mean}_{\boldsymbol{p}\in\mathbb{P}}\left(\min_{\boldsymbol{p}^*\in\mathbb{P}^*}\|\boldsymbol{p}-\boldsymbol{p}^*\|_2\right)$ |
| Completeness | $\text{mean}_{\boldsymbol{p}^*\in\mathbb{P}^*}\left(\min_{\boldsymbol{p}\in\mathbb{P}}\|\boldsymbol{p}-\boldsymbol{p}^*\|_2\right)$ |
| Chamfer | $(\text{Accuracy} + \text{Completeness})/2$ |
| Precision (%) | $\text{mean}_{\boldsymbol{p}\in\mathbb{P}}\left(\min_{\boldsymbol{p}^*\in\mathbb{P}^*}\|\boldsymbol{p}-\boldsymbol{p}^*\|_2 < 0.05\right)\times 100$ |
| Recall (%) | $\text{mean}_{\boldsymbol{p}^*\in\mathbb{P}^*}\left(\min_{\boldsymbol{p}\in\mathbb{P}}\|\boldsymbol{p}-\boldsymbol{p}^*\|_2 < 0.05\right)\times 100$ |
| F-score (%) | $2\times\text{Precision}\times\text{Recall}/\left(\text{Precision}+\text{Recall}\right)$ |

(b) Evaluation Metrics for Segmentation.

| Metric | Definition |
|---|---|
| Rand Index | $\dfrac{N_{\mathbb{C}}(N_{\mathbb{C}}-1)-2N_{\text{disagree}}(\mathbb{C},\mathbb{C}^*)}{N_{\mathbb{C}}(N_{\mathbb{C}}-1)}$ |
| Variation of Information | $H(\mathbb{C})+H(\mathbb{C}^*)-2I(\mathbb{C},\mathbb{C}^*)$ |
| Segmentation Covering | $\dfrac{C(\mathbb{C}^*\to\mathbb{C})+C(\mathbb{C}\to\mathbb{C}^*)}{2}$ |

1 to 128 using F-score, VOI, and SC metrics on ScanNetv2 dataset. As illustrated in Fig. 10, the performance of NeuralPlane is almost optimal when $d$ is larger than 2. Similar results are also observed by ContrastiveLift (Bhalgat et al., 2023) and AirPlanes (Watson et al., 2024). In light of this, we simply choose $d = 4$ in our experiments for a trade-off between performance and efficiency.

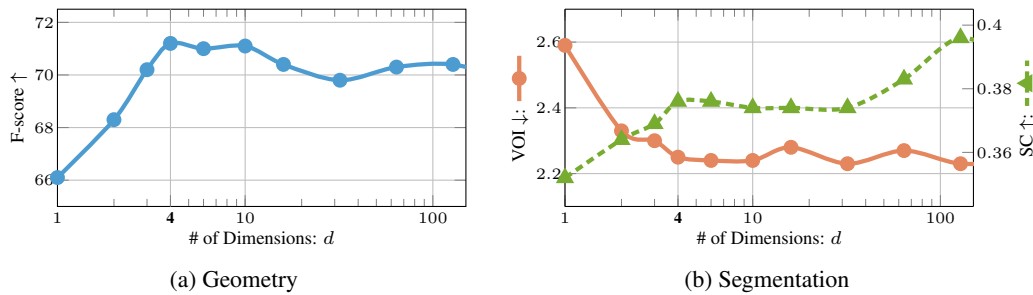

(a) Geometry         (b) Segmentation

Figure 10: **Impact of different numbers of feature dimension.**

**Ablating DBSCAN Epsilon.** To supervise the neural parser module, our proposed method uses DBSCAN to first cluster the noisy primitive-level features rendered during each iteration. The study on the impact of varying the DBSCAN epsilon is presented in Fig. 11. The results indicate that the performance of NeuralPlane is close to optimal when the epsilon is in the range of 0.2 to 0.5. A higher cluster epsilon leads to coarse-grained feature discrimination, while a lower epsilon may result in over-segmentation. However, thanks to the robust mechanism of point grouping and parameter estimation introduced in Sec. 3.3, no significant performance deterioration is observed even if DBSCAN is ineffective (with the cluster epsilon set to 2).

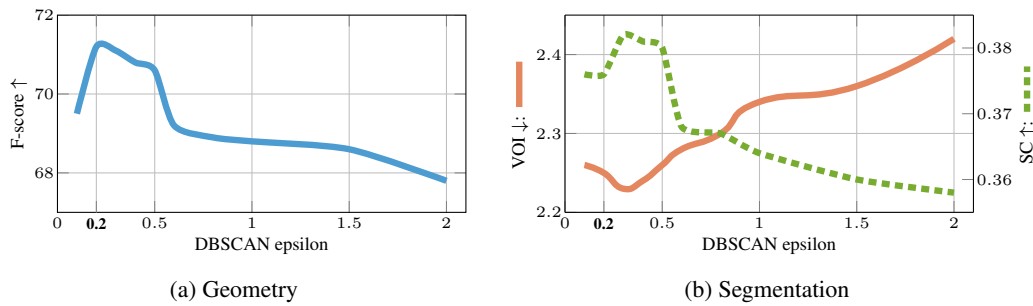

(a) Geometry         (b) Segmentation

Figure 11: **Performance on ScanNetv2 with varying DBSCAN epsilon.**

**A Sensitivity Analysis of SfM Geometry.** Incorporating geometric cues, such as estimated monocular depth and normal maps, into neural scene representations is widely adopted to improve the performance of indoor 3D surface reconstruction (Yu et al., 2022b; Wang et al., 2022; Park et al., 2024). Likewise, NeuralPlane requires additional geometric priors for the initialization of local planar primitives. We find that, although sparse and noisy, the readily available SfM point cloud

is sufficient to provide a good initialization. To further analyze the robustness of our approach to SfM geometry of different quality levels, we utilize different SfM point clouds triangulated from correspondences estimated with various feature matching techniques. Results in Tab. 6 demonstrate that dense matching methods, *e.g.,* LoFTR (Sun et al., 2021), are more favorable for our method, as sparse matching methods, *e.g.,* SuperPoint (DeTone et al., 2018) + LightGlue (Lindenberger et al., 2023), are struggling with low-texture regions. Besides, our approach can consistently benefit from the development of more advanced feature matcher, *e.g.,* RoMa (Edstedt et al., 2024).

Table 6: **Performance of NeuralPlane initialized with different SfM keypoints.**

| Method | Geometry | | Segmentation | | |
|---|---|---|---|---|---|
| | Chamfer ↓ | F-score ↑ | RI ↑ | VOI ↓ | SC ↑ |
| w/o SfM Geometry | 10.13 | 40.2 | 0.925 | 3.08 | 0.293 |
| SuperPoint + LightGlue | 5.90 | 65.4 | 0.942 | 2.49 | 0.345 |
| LoFTR (Indoor) | **4.57** | 72.8 | **0.956** | 2.24 | 0.380 |
| LoFTR (MegaDepth) ← **in main paper** | 4.59 | 71.2 | 0.955 | 2.25 | 0.376 |
| RoMa (Indoor) | 4.62 | **73.2** | **0.956** | **2.12** | **0.400** |

**Evaluation on Reconstruction Efficiency.** In contrast to learning-based MVS methods that can be implemented online at interactive speeds, our proposed NeuralPlane is a relatively time-consuming method that requires per-scene optimization. However, our method is much more efficient than other offline neural surface reconstruction methods. Tab. 7 presents a comparison of average execution time against baseline models on ScanNetv2.

Table 7: **Comparison of time consumption across neural implicit methods.**

| | Pre-processing (min) | Training (h) | Inference (min) |
|---|---|---|---|
| ManhattanSDF (Guo et al., 2022) | 44.5 | 5.6 | 2.3 |
| NeuRIS (Wang et al., 2022) | 0.5 | 4.2 | 1.2 |
| MonoSDF (MLP) (Yu et al., 2022b) | **0.3** | 7.5 | 0.7 |
| NeuralPlane | 2.9 | **0.1** | **0.5** |

**Additional Qualitative Results.** In Figs. 12 and 13, we visualize the entire 3D plane reconstruction on ScanNetv2 and ScanNet++ datasets, drawing comprehensive comparisons against state-of-the-art methods. Moreover, we provide close-up views rendered from reference poses in Fig. 14.

**Out-of-Domain Experiment.** While our work primarily focuses on indoor scenarios, we also conducted experiments on two small-scale outdoor scenes from the Niantic MapFree dataset (Arnold et al., 2022). The results in Fig. 15 highlight the potential of our method to generalize to outdoor environments.

## A.5 LIMITATIONS AND FUTURE WORK

**Failure Cases.** Fig. 16 illustrates several common failures, showcasing issues such as (a) missing structures and (b) under-segmentation, both of which stem from inaccuracies in 2D plane observations. The method is also not equipped to handle (c) non-planar surfaces. Besides, manually setting the number of semantic prototypes limits the adaptability of our method; for instance, an excessively large value may lead to (d) over-segmentation.

NeuralPlane at its core is integrating noisy 2D plane observations into a unified 3D neural representation, but it may fail to recover from catastrophic errors such as severe inaccuracy in mono-normal estimation and SfM geometry, which we did not consider. Meanwhile, the method is currently restricted to compact environments and better suited to indoor settings. Complex and large-scale scenes continue to pose many challenges, including (1) the need for large model capacities, and (2) the presence of massive non-planar clutter, which we leave for future research. Intersection analysis (Nan & Wonka, 2017) or integrating other cues such as occlusion edges for more structured reconstruction is another interesting research topic. It is also promising to implement our method in an online and efficient SLAM framework (Zhu et al., 2022; Matsuki et al., 2024).

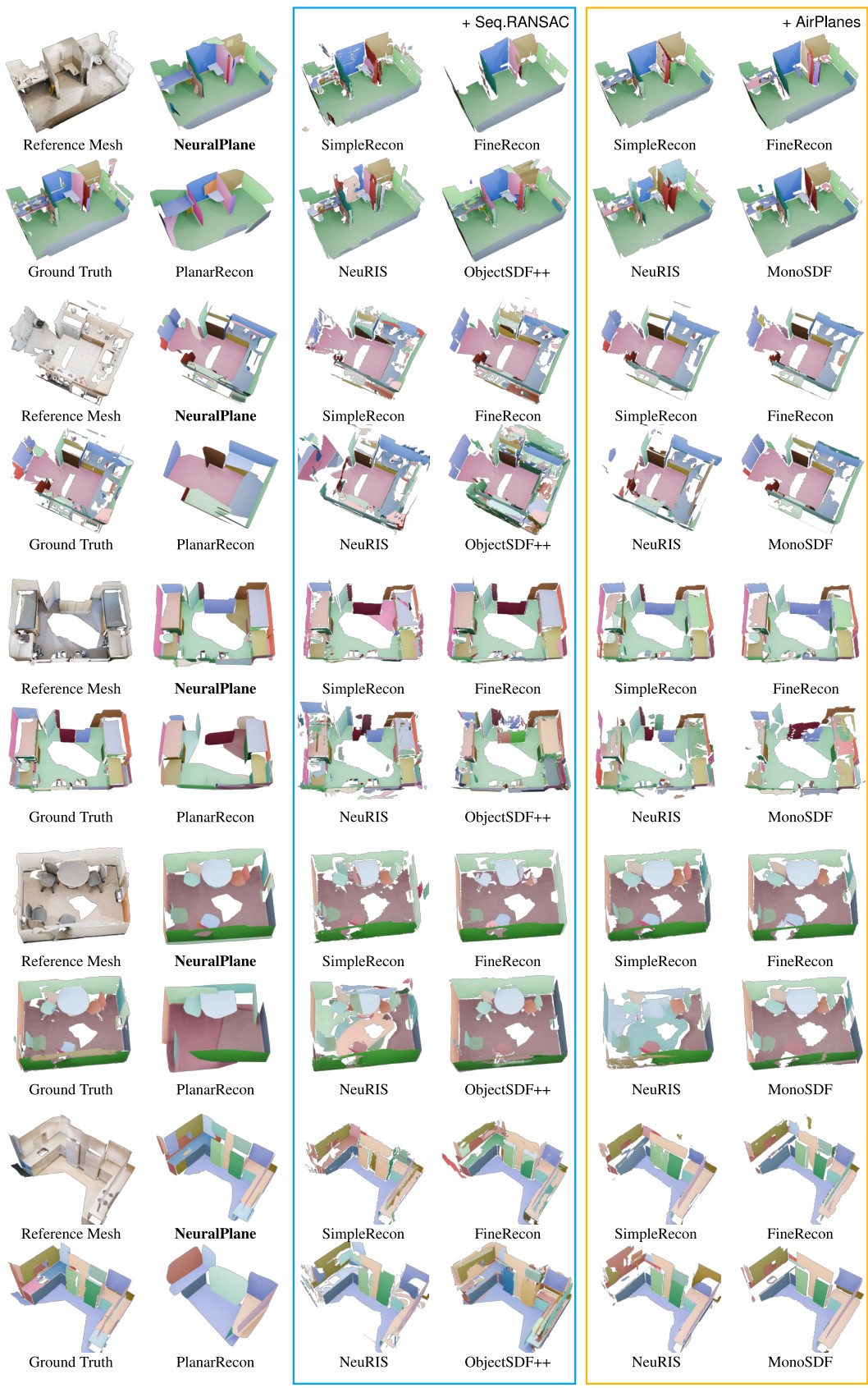

Figure 12: **More qualitative comparisons on ScanNetv2.**

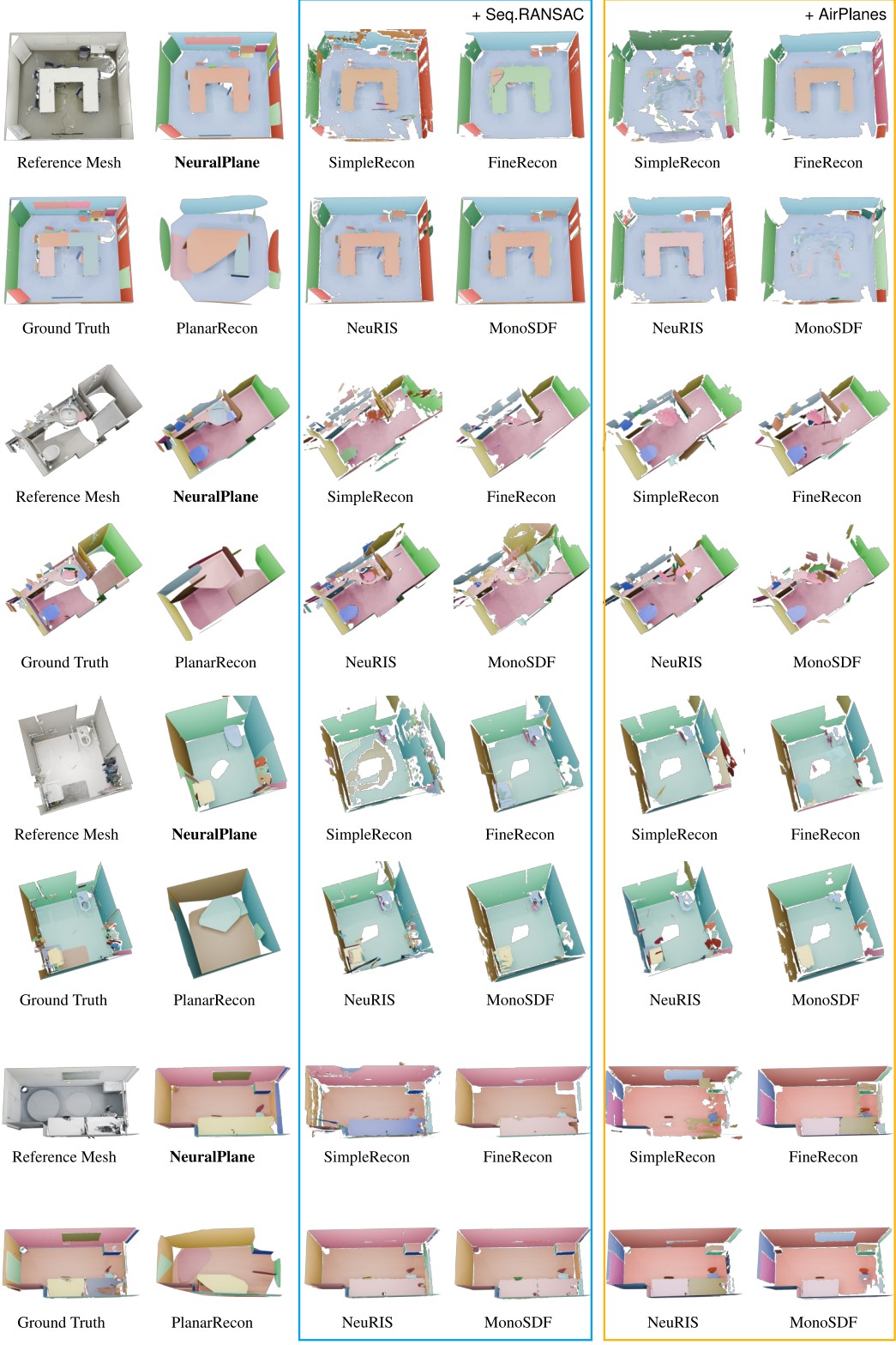

Figure 13: **More qualitative comparisons on ScanNet++.** Ceilings and occluding walls have been filtered for better visualization.

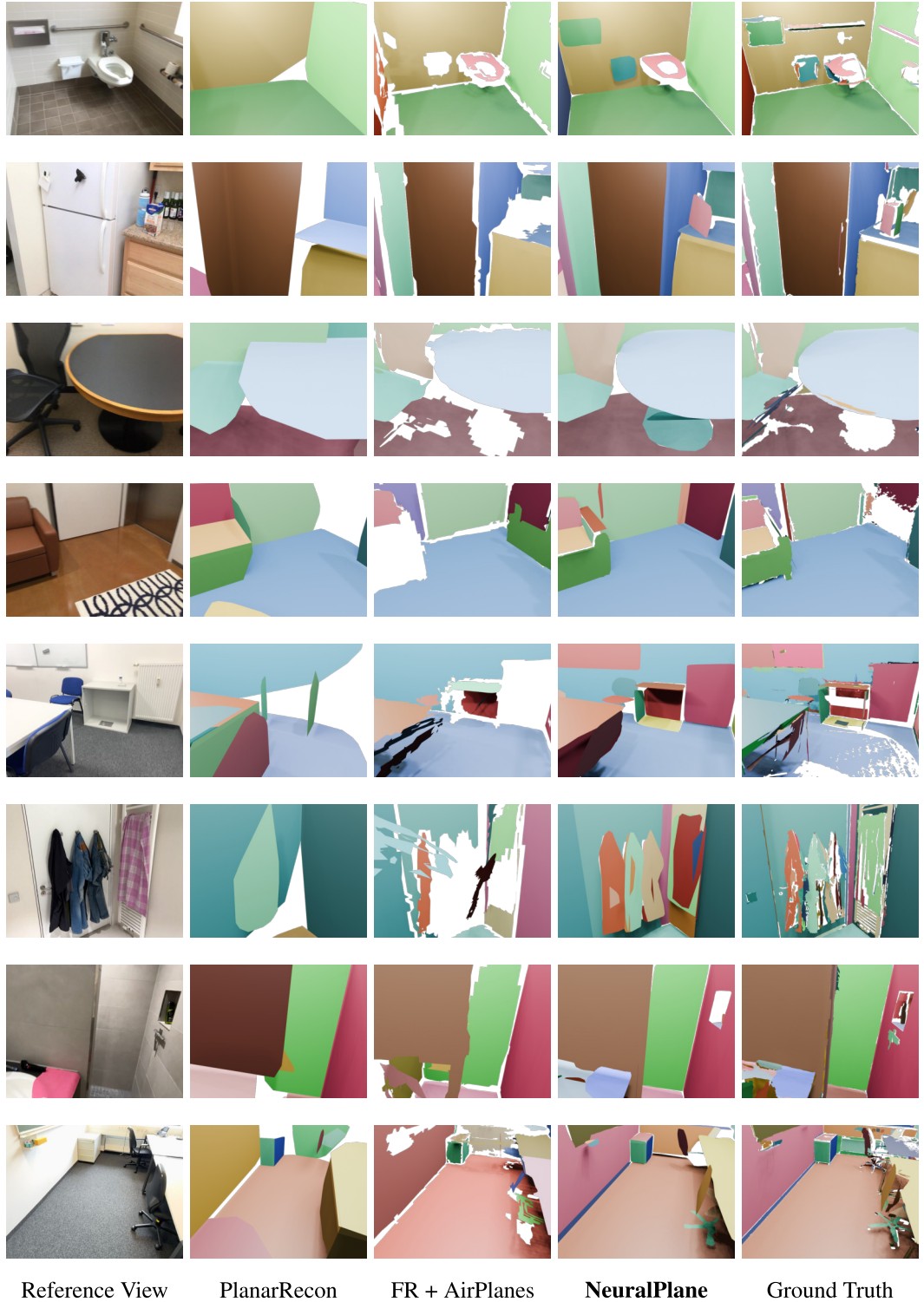

Reference View    PlanarRecon    FR + AirPlanes    **NeuralPlane**    Ground Truth

Figure 14: **Close-ups of qualitative results.**

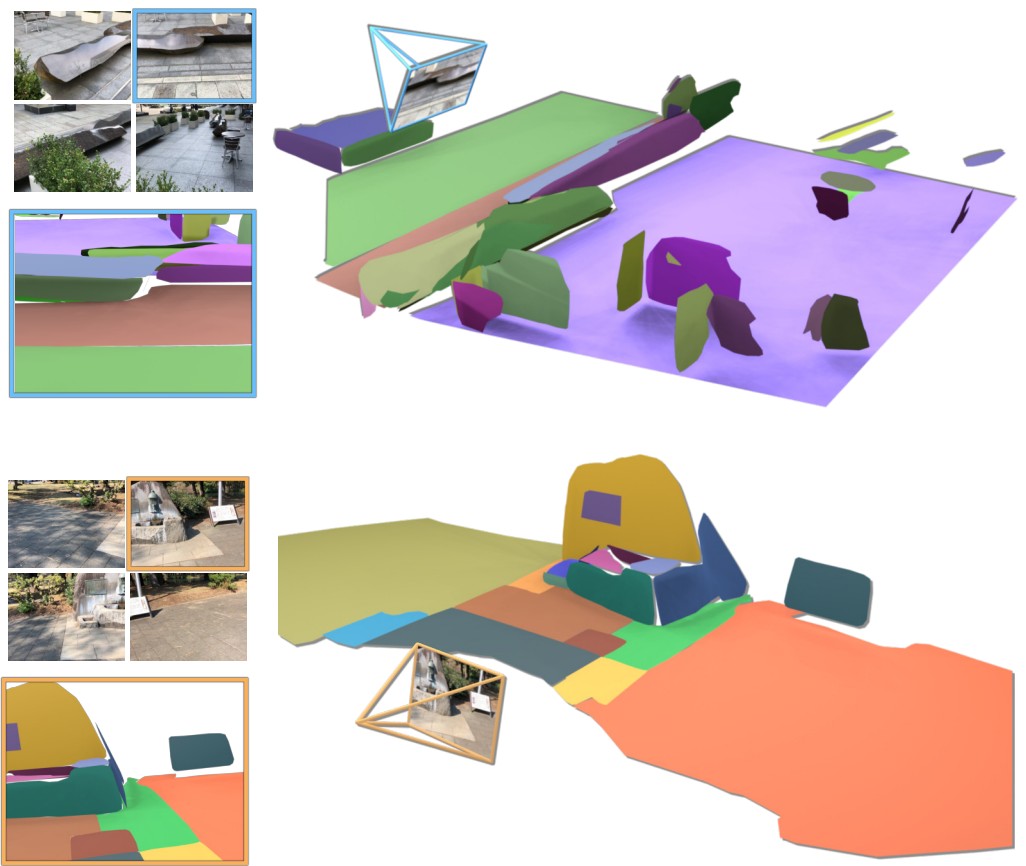

Figure 15: **Our method is applicable to outdoor environments.** As no ground-truth reconstruction is available, we recommend viewing videos in the supplementary for a more comprehensive evaluation.

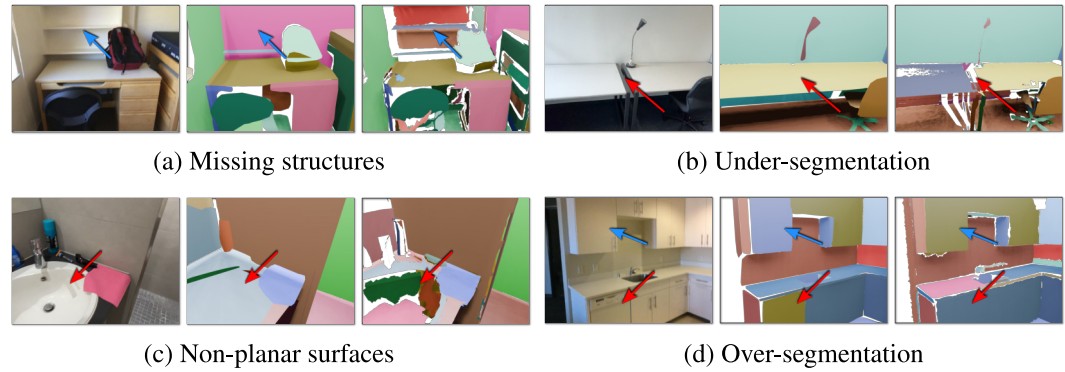

(a) Missing structures        (b) Under-segmentation

(c) Non-planar surfaces        (d) Over-segmentation

Figure 16: **Failure Cases.** In each case, from left to right: reference image, our reconstruction, and the ground truth. (a) Planes nearing the optical axis, as well as intricate structures, are difficult to detect in 2D and are thus likely to be absent from the final reconstruction. (b) The failure to semantically differentiate between two adjacent planes in certain views could lead to under-segmentation. (c) Non-planar surfaces are forcibly fitted into planes. (d) Over-segmentation due to a too large value of semantic prototypes.

