# OpenReview forum: "NeuralPlane: Structured 3D Reconstruction in Planar Primitives with Neural Fields"
_ICLR.cc/2025/Conference — ICLR 2025 Oral_

### Official Review · Reviewer_YwVB · 2024-10-26

**Soundness:** 3
**Presentation:** 3
**Contribution:** 3
**Rating:** 8
**Confidence:** 4

**Summary:**

This paper proposed a framework called NeuralPlane to reconstruct 3D indoor scenes as planar primitives from posed 2D images. The author first employs 2D prior models to generate local planar primitives, then uses the geometric and semantic priors to guide the NeRF-style reconstruction learning. Finally, a decoding algorithm is designed to extract the global explicit plane mesh from the learned neural field.
Extensive experiments are conducted to evaluate the performance on ScanNetv2 and ScanNet++ datasets.

**Strengths:**

This paper proposed a comprehensive 3D reconstruction framework with plane primitives.

* Compared to some of the similar works that only focus on detecting the planes, this work also utilizes the detected planes to guide neural field learning.

* The presentation of this paper is clear and concise.

* The qualitative and quantitative results look promising.

**Weaknesses:**

* This work involves a lot of submodules, especially the neural field with geometric, semantic, and coplanar features could be computationally expensive.

* A concurrent work is very relevant to this paper and could be discussed in the related works section:
Chen, Zheng, et al. "PlanarNeRF: Online Learning of Planar Primitives with Neural Radiance Fields." arXiv preprint arXiv:2401.00871 (2023).

* The text illustration and conceptual figure of Neural Parser could be improved, current version is not clear enough and easy to follow.

**Questions:**

* What's the computation complexity of this work? Like the GPU memory usage and training time? How's it compared to related works?

---

> ### Author Response · Authors · 2024-11-20
> **Comment to Official Review**
>
> We thank the reviewer YwVB for the constructive feedback and valuable suggestions!
> Below, we provide a detailed response to your questions and comments.
>
> > **W1: Computational complexity.**
>
> We appreciate the reviewer's concern about the complexity of our method.
> Acknowledging this common concern raised by all reviewers, **in our general response above**, we have presented additional experimental results on the complexity of the methodology,
> where we measured the average training duration and peak GPU memory usage for each of the proposed modules, and compared our work with other methods.
> Generally speaking, our method is **computationally inexpensive and efficient** when compared to existing neural implicit reconstruction methods.
>
> To clarify: compared to the learning-based methods such as *PlanarRecon* and *AirPlanes*, our method is relatively time-consuming that requires per-scene optimization.
> However, we deem that the neural implicit method is more generalizable (please also refer to the out-of-domain experiment section in our general response).
> Besides, there are numerous SLAM frameworks based on neural representations (or say, differentiable rendering), such as *NICE-SLAM* and *Gaussian Splatting SLAM* which could enable us to implement our method in a more efficient and online fashion.
> We believe that this is an interesting direction for future research.
>
> > **W2: Missing a related work.**
>
> Thank you for the reference.
> _PlanarNeRF_ is indeed a highly relevant work that uses RGB-D sequences for dense 3D planar primitives detection in an online fashion.
> We will include it in the revision.
>
> > **W3: Illustration of Neural Parser needs improvement.**
>
> Thank you so much for pointing this out.
> Basically, this module learns a set of feature centroids in NCF during the training process, which is found to be effective in decomposing the scene into coplanar segments.
> We regret that, due to space constraints, we had to condense several points in the main text, drawing on a range of unexpanded prerequisites that may not be familiar to all readers.
> We will make rearrangements there to enhance readability.

---

> ### Author Response · Authors · 2024-11-24
> **A Gentle Reminder**
>
> Dear Reviewer YwVB,
>
> Thanks again for your efforts and suggestions for this paper.
> As the deadline for the author-reviewer discussion phase is nearing, we would like to courteously inquire if our responses have effectively addressed the concerns you raised.
> Should any remain unresolved, please let us know, and we will promptly follow up.
>
> Best regards, Authors

---

> > ### Comment · Reviewer_YwVB · 2024-11-25
> >
> > Dear Authors:
> >
> > Thank you for the detailed response, which addresses all of my concerns and is appreciated! I'll keep my rating.

---

> ### Author Response · Authors · 2024-11-27
> **Many thanks to the reviewer for their valuable feedback**
>
> Dear Reviewer YwVB,
>
> We sincerely appreciate your insightful comments and further prompt reply.
> We are glad to hear that the concerns have been addressed.
> Your recognition of our paper's strength and the clear acceptance are greatly valued.
>
> Thank you again for your involvement throughout the entire review process.
>
> Best regards, NeuralPlane Authors

---

### Official Review · Reviewer_1CZJ · 2024-11-03

**Soundness:** 4
**Presentation:** 4
**Contribution:** 4
**Rating:** 10
**Confidence:** 4

**Summary:**

The paper presents NeuralPlane, a novel approach to 3D plane reconstruction that utilizes neural fields for generating structured 3D maps from multi-view images without the need for plane annotations. The method emphasizes two main aspects: geometry and semantics. Key contributions include:
1. Monocular Plane Segmentation: A monocular module extracts geometrically smooth (based on off-the-shelves Surface Normal Predictor) and semantically meaningful 2D plane observations (based on Segmenet Anything Model).
2. Plane-Guided Neural Representation: The model utilizes these 2D segments to train a neural field that captures accurate 3D plane locations. A surface normal regularization and pseudo-depth regularization terms are proposed.
3. Neural Coplanarity Field: This self-supervised feature field enables semantic consistency within the 3D reconstruction by grouping planar regions that share coplanar relationships. A contrastive loss is proposed to distinguish between planes with similar geometric properties but different semantic properties.
The method demonstrates superior performance on ScanNetv2 and ScanNet++ datasets, indicating its effectiveness in indoor environments.

**Strengths:**

1. Novelty in Combining Geometry and Semantics: The method’s approach to merging geometry with semantic information through a neural coplanarity field is innovative, enhancing the semantic consistency of 3D reconstructions. A complex system with multiple stages is proposed to estimate/reason the local planar regions and the associated parameters, to associate the planes in 3D using radiance field, to resolve semantic conflicts using Neural Coplanarity Field.
2. High-Quality Reconstruction: Experimental results indicate that NeuralPlane achieves fine-grained and coherent plane reconstructions, outperforming existing methods in most of the metrics.
3. Efficiency: NeuralPlane’s volume density representation allows for faster training compared to implicit methods, an important practical advantage.
4. Extensive ablation studies: the authors include sufficient ablation studies to support the effectiveness of the proposed modules.
5. Good writing: the paper is clearly written that the technical details are clearly presented.

**Weaknesses:**

1. Complexity of Methodology: The proposed method involves several stages, including monocular plane segmentation, neural coplanarity field training, and plane extraction. In each stage, there are several submodules and I believe there are some hyperparameters decisions in each stage. While effective, this complexity (especially the combination of submodules and associated parameters) may impact its scalability to larger scenes or generalizability. For example, K-means clustering on predicted normal map, mask size threshold in SAM, thresholds to form negative pairs in Neural Coplanarity Field, Loss balancing parameters, etc. It would be great if the authors could share the insights on the impact of these hyperparameter settings on different scenes. For example, is a universal set of hyperparameters applied to all the test scenes? It would be great if the authors could provide a sensitivity analysis or ablation study on key hyperparameters across different scenes. This would help clarify how robust the method is to parameter changes and whether a universal set of parameters is feasible.

2. Dependence on Initial 2D Plane Segmentation Quality: As the regularizations are based on the quality of initial local plane geoemtry, the method’s success can depend on the quality of 2D plane segments obtained from monocular priors, which may introduce inaccuracies in challenging environments for monocular predictors. As the authors mentioned , the local planar primitives can result in severe inconsistency across views (Line 194).

3. Over-Segmentation Issue: As highlighted in the paper, the Segment Anything Model (SAM) tends to over-segment planes, resulting in multiple smaller plane segments for a single surface. Although this is managed in the training process, it may require further refinement to avoid segmentation inconsistencies in complex scenes. From the visualization on the GitHub page, it seems to me that some of the small segments usually resulted in inaccuracy.

**Questions:**

1. Eqn 4: n_i is not defined, is it the surface normal of the selected local planar primitive? Please explicitly define n_i in the text or equation, and confirm if it refers to the surface normal of the local planar primitive.

**Details Of Ethics Concerns:**

/

---

> ### Author Response · Authors · 2024-11-20
> **Comment to Official Review**
>
> We'd like to thank the reviewer for their time in providing a detailed review with insightful comments and for their recognition of our work.
>
> > **W1: Complexity of methodology.**
>
> We appreciate the reviewer's concern about the complexity of our method.
> Acknowledging this common concern raised by all reviewers, **in our general response above**, we have presented additional experimental results on the complexity of our method and its robustness to hyperparameters.
> Generally speaking, our method is **computationally inexpensive** and **robust to most of the concerned hyperparameters**, with the performance close to the optimum when they are set within a reasonable range.
> Please be aware that in our main experiment, **a universal set of hyperparameters is applied to all test scenes** and we will explicitly clarify this in the revision.
>
> Regarding the K-means clustering on predicted normal map, we referred to PlaneRCNN and PlanarRecon, which respectively utilize 7 and 6 normal anchors to estimate plane normals.
> We make a difference by choosing K=8 and reserving only the 6 principal clusters, which empirically leads to improved results.
> Furthermore, as you mentioned, we filter out noisy 2D plane segments based on their pixel areas.
> All these efforts are to ensure the salience of the detected 2D planes.
> Sorry that at the present time, we do not have conclusive data on how to determine these values according to various scenes.
> We will continue to investigate this.
>
> We need to clarify that the method is currently restricted to compact environments and better suited to indoor settings.
> Although our method is applicable to small outdoor scenes,
> large-scale and complex outdoor scenes continue to pose many challenges, including (1) the need for large model capacities, and (2) the presence of massive non-planar and spurious clutter, which we will leave as future research.
>
> > **W2: Dependence on the quality of initial local planar primitives.**
>
> The reviewer's observation is valid, and the limitation is also discussed in *Appendix A.5*, where we note that our method may fail to recover from catastrophic errors in such as severe inaccuracy in mono-normal estimation and SfM geometry.
>
> Meanwhile, following the reviewer FCSv's suggestion, we additionally conduct an out-of-domain experiment on two small-scale outdoor scenes.
> The results indicate that by repurposing the pretrained monocular normal predictor and the Segment Anything Model as 2D foundational models, the proposed method has the potential of generalizing to challenging environments.
> We invite the reviewer to check our updated supplementary material for these qualitative results.
>
> > **W3: Over-segmentation issue.**
>
> We value the reviewer's insightful feedback.
> You are right that, although the over-segmentation issue in 2D could largely be managed during the training process, there are still cases where undesirable over-segmentation may occur in the final reconstruction, such as where the floor could be divided into multiple pieces.
> We partially attribute this to the inherent ambiguity in determining a semantically coherent plane structure and will further investigate to handle this semantic ambiguity.
>
> Regarding the small segments you concerned, we observe that they mostly arise from the noisy 2D plane segments.
> These segments usually correspond to non-planar clutter and can only be detected in a limited number of views, which could lead to the inaccuracies.
> We plan to tackle these limitations in the future.
>
> > **Q1: $\hat{\mathbf{n}}_i$ in Eq. 4 is undefined.**
>
> We apologize for this confusing notation.
> Firstly, the $\hat{\mathbf{n}}$ is defined before as the **NeRF-derived** normal, rather than the normal of the local planar primitive $P$ which is actually denoted as $\bar{\mathbf{n}}$.
> The NeRF-derived normal $\hat{\mathbf{n}}$ is estimated from a randomly sampled ray triplet in an algebraic manner (*Eq. 3*).
> Then, to compute the normal loss $\mathcal{L}_{\text{normal}}$ (*Eq. 4*), multiple ray triplets will be sampled, so that the subscript $i$ added to $\hat{\mathbf{n}}$ denotes the estimated NeRF-derived normal of the **i-th** sampled ray triplet $\mathcal{T}_i$.
> We will clarify this in the revision.

---

> > ### Comment · Reviewer_1CZJ · 2024-11-22
> > **reply to author response**
> >
> > Thank you for providing the additional experiments and results in your response. They address my concerns and clarify the issues I raised. I will maintain my original rating for this submission.

---

> ### Author Response · Authors · 2024-11-27
> **Many thanks to the reviewer for their valuable feedback**
>
> Dear Reviewer 1CZJ,
>
> We sincerely appreciate your insightful comments and further prompt reply.
> We are glad to hear that the concerns have been addressed.
> Your recognition of our paper's strength and strong recommendation are a great encouragement to us.
>
> Thank you again for your involvement throughout the entire review process.
>
> Best regards, NeuralPlane Authors

---

### Official Review · Reviewer_6wEq · 2024-11-03

**Soundness:** 3
**Presentation:** 3
**Contribution:** 3
**Rating:** 6
**Confidence:** 3

**Summary:**

This paper presents NeuralPlane, a method for reconstructing 3D scene plane primitives via neural fields without GT plane labelling. The method is divided into three main stages: firstly, it combines pre-trained normal prediction and SAM to generate initial 2D planar segments and estimates their 3D parameters using SfM keypoints. Secondly, it optimises two neural fields, a density field based on planar geometric constraints, and a coplanar neural field that understands the semantic relationships between regions . The neural coplanar field is followed by a neural parser module that helps to model the learned coplanar relations. Finally, the optimised neural representations are converted into explicit 3D planes through point sampling, feature-based clustering and RANSAC fitting. The method is evaluated on the ScanNetv2 and ScanNet++ datasets, and the results show that it outperforms both the learning-based method and the Geometry+RANSAC method in terms of geometric and semantic metrics.

**Strengths:**

- This paper is well-written and easy to understand.
- The proposed method does not require ground-truth plane annotations, as it can learn effectively from noisy monocular model outputs.
- The method demonstrates SOTA performance and achieves clean plane segmentation results.

**Weaknesses:**

- The proposed method involves numerous hyperparameters, including balancing parameters for loss, the number of semantic prototypes, and parameters listed in Lines 850-863.
- As a complex system, it is important to discuss and present failure cases to help readers understand the method’s limitations.

**Questions:**

Since it is a complex system paper, making it hard to reproduce, will the code be publicly avaliable?

---

> ### Author Response · Authors · 2024-11-20
> **Comment to Official Review**
>
> Thank you for your motivating and positive feedback on our work.
> We provide a detailed response below to each of your concerns.
>
> > **W1: Numerous hyperparameters.**
>
> We agree with the reviewer's comment since the method involves a number of stages and submodules.
> Acknowledging this common concern raised by all reviewers, we have presented additional experimental results **in our general response above**, where the robustness to the concerned hyperparameters for loss balancing and RANSAC is further assessed.
> Generally speaking, our method is **robust to most of the concerned hyperparameters**, and its performance is close to the optimum when they are set within a reasonable range of values.
>
> Please also be aware that in our main experiment, a fixed set of hyperparameters is applied to all test scenes and we will explicitly clarify this in the revision.
>
> > **W2: Missing failure cases.**
>
> Thank you for this point.
> In the submission, we provide a brief discussion of the current limitations in *Appendix A.5*:
> (1) our method may fail to recover from catastrophic errors in 2D priors, such as severe inaccuracy in mono-normal estimation and SfM geometry;
> (2) the number of semantic prototypes has to be fixed or manually set, which potentially limits the scalability.
> Besides, we have to clarify that our method is currently restricted to compact environments and better suited to indoor settings.
>
> We will include further qualitative results of failure cases to help the reader have a thorough understanding.
>
> > **Q1: On code availability.**
>
> For the implementation complexity, we will release the code to support reproducible experiments and facilitate future research.

---

> > ### Comment · Reviewer_6wEq · 2024-11-25
> >
> > Dear Authors:
> >
> > Thank you for your reply and the hyperparameter experiments. Definitely, I agree with accepting the paper. Still, it is a very complex system with many hyperparameters. It has only been tested on a limited number of scenarios (only 12) and no visualizations of the failures have been provided, so I would like to keep my original rating.
> >
> > Best,
> > Reviewer 6wEq

---

> ### Author Response · Authors · 2024-11-24
> **A Gentle Reminder**
>
> Dear Reviewer 6wEq,
>
> Thanks again for your efforts and suggestions for this paper.
> As the deadline for the author-reviewer discussion phase is nearing, we would like to courteously inquire if our responses have effectively addressed the concerns you raised.
> Should any remain unresolved, please let us know, and we will promptly follow up.
>
> Best regards, Authors

---

> ### Author Response · Authors · 2024-11-27
> **Many thanks to the reviewer for their valuable feedback**
>
> Dear Reviewer 6wEq,
>
> We sincerely appreciate your insightful comments and further prompt reply.
> Your recommendation is a great encouragement.
>
> Following your constructive suggestion, we have included elaboration as well as visulizations of the common failure cases in _Appendix A.5_ (on page 20).
> Regarding the concern that only a limited number of scenarios were tested, in fact, we had carefully considered this aspect prior to our submission.
> However, we still opted to proceed with evaluating several representative scenes, noting that numerous studies on indoor scene reconstruction using neural implicit representations, such as _ManhattanSDF_ and _NeuRIS_, have adopted similar approaches due to efficiency considerations.
> Moreover, unlike feed-forward methods such as _PlanarRecon_, which are efficient but rely on 3D supervision, our method is annotation-free using 2D foundational models, and has exhibited superior robustness to outdoor scenarios (please refer to the **Out-of-domain experiment** in our general response).
>
> Finally, we acknowledge that your concern is completely valid: for an effective reconstruction system, it is essential to test across a wide range of scenarios to verify its robustness and investigate the relevant hyperparameters.
> We will continue to work on this in the future.
>
> Thank you again for your involvement throughout the entire review process.
>
> Best regards, NeuralPlane Authors

---

### Official Review · Reviewer_FCSv · 2024-11-04

**Soundness:** 3
**Presentation:** 3
**Contribution:** 3
**Rating:** 8
**Confidence:** 5

**Summary:**

This paper presents a neural 3D reconstruction system on 3D plane reconstruction of indoor scenes. Inspired by the recent success of neural radiance field and image foundation models (SAM2), this paper presents a multi-view 3D reconstruction pipeline leveraging these techniques. The training scheme consists of three phases: Initializing plane segments and parameters -> optimizing a neural feature field for plane-specific feature representation (encouraged by a list of geometry-guided loses) -> plane extraction by grouped features and RANSAC. Experiments are conducted extensively on two representative indoor datasets: ScanNet and 7-scenes and are compared with a group of competitve baseline methods.

**Strengths:**

1. The unique advantage of the system is the association of geometry and semantic features for plane reconstruction problem, which requires both geometry-aware perception and semantic-aware grouping.
2. The paper leverages foundation models to achieve plane segment initializaing and utilize geometry-driven losses to optimize the system. There is no groundtruth plane segmentation or geometry label required.
3. Overall the paper and diagrams are well written and presented.
4. The experimental comparison and thorough and convincing on both plane geometry reconstruction and segmentation.

**Weaknesses:**

1. I think the major advantage of this paper is the unsupervised learning paradigm on the plane reconstruction problem. However, since both ScanNet and ScanNet++ has groundtruth images, this unique advantages seems not to be fully enjoyed and reflected. So I suggest authors to apply the proposed system on some outdoor scenes containing plane structures (such as autonomous driving dataset or street view datasets), to verify the adaptability of the method.

2. I have tested PlaneRecon in my previous projects. It can incrementally reconstructs planes in an online and real-time manner. Besides, it can trains over multiple scenes and directly test without any test-time optimization. Although its precision should fall behind than the neural reconstruction papers, the generalizability and speed is higher than the proposed method. I am curious on whether the paper has such potential to tackle these limitations.

3. Missing a few related works: (1) Recovering 3d planes from a single image via convolutional neural networks (2) PlaneMVS: 3D Plane Reconstruction From Multi-View Stereo (3) Single-image piece-wise planar 3d reconstruction via associative embedding.

4. On quantitative evaluation, it is unclear that what is the groundtruth is here. Does it stand for the groundtruth planar part only or the entire mesh? As a plane reconstruction paper, I think the former one should be more reasonable. If so, the first several methods listed in Table 1 cannot directly be compared with the proposed method. Authors should make it clearer on this part.

5. Is the proposed method robust to the hyperparameters listed in the paper especially (1) to,tn and m in the push loss during training and (2) the parameters selected in RANSAC during plane fitting? Some ablation studies on hyper-parameter robustness are expected to make the method more generalizable across most indoor scenes, since the geometric scale and semantic distribution can have large variance among scenes.

**Questions:**

I am impressed by the technical contribution made by authors for this work. However, there also exist a few major concerns for me at this time which discourage me to grant this paper a higher value.

Please try to address my concerns listed in the weakness part. I will accordingly consider to improve my overall rating if the concerns are well solved.

---

> ### Author Response · Authors · 2024-11-20
> **Comment to Official Review (1/2)**
>
> We appreciate the reviewer's thoughtful feedback and valuable suggestions.
> Below, we provide further clarification and details to your concerns.
> If any of our responses do not fully address your concerns, please let us know, and we will promptly follow up.
>
> > **W1: Evaluations on some outdoor scenes, verifying the adaptability of the method.**
>
> This is a noteworthy experiment that we had not investigated prior to submission.
> Our work and recent studies on 3D plane reconstruction primarily focus on indoor scenarios, but honestly, we too have wondered about the adaptability of our method to outdoor scenes.
> To this end, we select two outdoor scenes from the training split of the [Niantic MapFree dataset](https://research.nianticlabs.com/mapfree-reloc-benchmark).
> Each scene depicts a small outdoor place and comes with two independent scans.
> Only the first scan is utilized for the experiments.
> We consider this to be a completely out-of-domain test,
> as no existing method has been specifically designed to handle such scenes.
> All settings used in this experiment are kept consistent with those presented in our original submission.
>
> Since no ground-truth reconstruction is available for these scenes, we encourage the reviewer to **refer to our updated supplementary material** for detailed qualitative results.
> The results demonstrate the robustness of our method and its potential to generalize to challenging outdoor environments.
> Here, we need to point out that *PlanarRecon* collapsed on both scenes, while geometry+RANSAC baselines employing learning-based MVS reconstruction methods, such as *SimpleRecon* and *DoubleTake*, failed to yield meaningful results.
>
> > **W2: The proposed method is time-consuming which needs test-time optimization.**
>
> *PlanarRecon* is the first learning-based model that we highly appreciate for its real-time global 3D plane reconstruction capability.
> The reviewer's concern about the time efficiency of our method is **indeed a valid point**.
> However, as a neural implicit approach, our work primarily aims at maintaining a consistent 3D representation of environmental plane structures, thus ensuring the quality of the final plane reconstruction.
> We regard this as an interesting exploration of neural fields for parametric primitives.
> More importantly, we advocate for the use of this powerful neural 3D representation, fusing various 2D observations from pretrained or foundational models.
> To alleviate the concerned efficiency limitation, we consider it is possible to integrate our method into a more efficient and online SLAM framework based on neural representations (or say, differentiable rendering), such as *NICE-SLAM* and *Gaussian Splatting SLAM*.
>
> Regarding the concern of generalizability, as discussed in **W1**, we have conducted an out-of-domain experiment in outdoor scenarios to verify the adaptability of our method, where learning-based feed-forward reconstruction methods collapse or fail to deliver meaningful results.
>
> > **W3: Missing a few related works.**
>
> Thanks for highlighting these references.
> They are representative prior works in reconstructing 3D plane from a limited number of views, and we will cite them in the literature review.

---

> ### Author Response · Authors · 2024-11-20
> **Comment to Official Review (2/2)**
>
> > **W4: The ground truth used for quantitative evaluation is unclear.**
>
> Sorry for the confusion.
> Following *PlanarRecon*, we use 3D planes, i.e., **only the ground-truth planar parts** provided by *PlaneRCNN*, as ground truth for evaluation.
> While in the first 7 rows of *Tab. 1*, we aim to assess the capability of state-of-the-art surface reconstruction methods, with the ground truth for their RAW geometry outputs being the official scene reconstruction (i.e., the entire meshes).
> Please note that, here, we do not intend to directly compare with these surface reconstruction methods, but rather to provide a quantitative demonstration of how the geometry+RANSAC paradigm relies on the quality of input geometries.
> To avoid any confusion, we will explicitly clarify this distinction in the main paper.
>
> > **W5: Robustness to hyperparameters.**
>
> Acknowledging this common concern raised by all reviewers, we have presented additional experimental results **in our general response above**, where the robustness of our method to hyperparameters in $\mathcal{L}_{\text{pull}}$ and RANSAC is assessed.
> Generally speaking, our method is **robust to most of the concerned hyperparameters**, and its performance is close to the optimum when they are set within a reasonable range of values.
>
> Please be aware that in our main experiments presented in submission, a fixed (i.e., universal) set of hyperparameters is applied to all test scenes.
> We also need to clarify that the method is currently restricted to compact environments and better suited to indoor settings.
> Although our method is currently applicable to small outdoor scenes,
> large-scale and complex outdoor scenes continue to pose many challenges, including (1) the need for large model capacities, and (2) the presence of massive non-planar and spurious clutter, which we plan to tackle in future research.

---

> > ### Comment · Reviewer_FCSv · 2024-11-21
> > **Feedback to authors' rebuttal**
> >
> > I appreciate the comprehensive and careful feedback from authors, as well as the supplemented experiments. The experiments and clarification look convincing and clear to me, especially the qualitative visualization on outdoor data to validate the generalizability of this method on zero-shot scenes. I will improve the overall rating and undoubtedly, the proposed method is novel, technically sound, and is in a good shape for acceptance. I am also looking forward to the code release of this paper for future research and exploration.

---

> ### Author Response · Authors · 2024-11-27
> **Many thanks to the reviewer for their valuable feedback**
>
> Dear Reviewer FCSv,
>
> We sincerely appreciate your insightful comments and further prompt reply.
> We are highly encouraged by your acknowledgment of our comprehensive response with new experiments.
> Specifically, following your constructive suggestion, we have included additional outdoor results in the appendix (on page 20 and 24).
>
> Thank you again for your involvement throughout the entire review process.
>
> Best regards, NeuralPlane Authors

---

### Author Response · Authors · 2024-11-20
**General Response by Authors**

We are deeply grateful to the reviewers for dedicating their time and effort to the reviewing process.
We are pleased to note that the reviewers find the presentation of the paper clear and concise (FCSv, 6wEq, 1CZJ, YwVB), and acknowledge the novelty of our work in associating geometry and semantics (FCSv, 1CZJ, YwVB) as well as delivering high-quality reconstruction without requiring plane annotations (FCSv, 6wEq, 1CZJ, YwVB).

We also notice that the reviewers' major concerns mainly lie in the **complexity of methodology** (6wEq, 1CZJ, YwVB) and its **robustness to hyperparameters** (FCSv, 6wEq, 1CZJ, YwVB).
Accordingly, during the rebuttal period, we measured the average training time and peak GPU memory consumption of each proposed module, and
conducted more detailed ablation studies on the ScanNetv2 dataset, with a particular focus on the analysis of hyperparameter sensitivity.
These additional measurements and experimental results presented below serve as general responses to the reviewers' shared concerns.
We hope these efforts will comprehensively evaluate the performance of our work and provide further clarity.

---

> ### Author Response · Authors · 2024-11-20
> **1. Complexity of the methodology**
>
> As listed below, we first report the profiling results for average training time and peak memory costs of each proposed module.
> Compared to the base model (i.e., *Nerfacto*), our full model approximately incurs a 1$\times$ increase in time consumption and a 15.8% increase in peak memory consumption.
> However, please note that only 4k iterations are required for training, adding only 2-3 minutes to the total time consumption, which could be considered acceptable.
>
> |                                                                    |$\text{\quad}$Average Training Time / 10iters $\text{\quad}$                                  | $\text{\quad}$Peak GPU Memory Usage (GB) $\text{\quad}$                                      |
> | :----------------------------------------------------------------- | :---------------------------------------------------------------- | ---------------------------------------------------------------- |
> | Nerfacto (the base model)                        | $\text{\quad}$303.6 ms                                         | $\text{\quad}$1.46                                                             |
> | $+\text{ }\mathcal{L}_{normal}$ (defined in *Eq. 4*)              | $+\text{ }$ 85 ms ( $\uparrow$ 28% )                              | $+\text{ }$ 0 ( $\rightarrow$ )                                  |
> | $+\text{ }\mathcal{L}_{p-depth}$ (defined in *Eq. 5*)$\text{\quad\quad}$ | $+\text{ }$ 26 ms ( $\uparrow$ 9% )                   | $+\text{ }$ 0.04 ( $\uparrow$ 2.9% )                 |
> | $+$ Refine.                                       | $+\text{ }$ 44 ms ( $\uparrow$ 14.5% )           | $+\text{ }$ 0.01 ( $\uparrow$ 1% )              |
> | $+$ NCF                                  | $+\text{ }$ 105 ms ( $\uparrow$ 34.7% )     | $+\text{ }$ 0.166 ( $\uparrow$ 11.4% )     |
> | $+$ NP                                  | $+\text{ }$ 71 ms ( $\uparrow$ 23.3% ) | $+\text{ }$ 0.001 ( $\uparrow$ 0.3% ) |
> | **Full Model**                                   | $\text{\quad}$**635.5 ms** ( $\uparrow$ 109.3% )                                         | $\text{\quad}$**1.69** ( $\uparrow$ 15.8% )                                           |
> |                            |        |               |
>
> In contrast to the learning-based MVS baselines, such as *PlanarRecon* and *AirPlanes*, which can be implemented at interactive speeds, *NeuralPlane* is currently a time-consuming method that requires offline optimization for each scene.
> It is a common issue for neural implicit reconstruction methods, but when compared to the state-of-the-art methods in this literature, as reported below, our method is significantly more efficient.
>
> |                            | NeuRIS$\text{\quad}$ | MonoSDF (MLP)$\text{\quad}$ | NeuralPlane (Ours) |
> | :------------------------- | :----- | :------------ | :----------------- |
> | Training Time (h)          | 4.2    | 7.5           | 0.1                |
> | Peak GPU Memory Usage (GB)$\text{\quad}$ | 8.0    | 4.7           | 1.7                |
> |                            |        |               |                    |

---

> ### Author Response · Authors · 2024-11-20
> **2. Robustness to hyperparameters (1/2)**
>
> Firstly, we would like to emphasize here and will explicitly clarify in the revision that in our main experiments presented in submission, a fixed (i.e., universal) set of hyperparameters is applied to all test scenes.
> In addition to the ablation studies we have conducted in *Sec. 4.3* and *Appendix A.4* which include the number of semantic prototypes, the feature dimension of NCF and the DBSCAN epsilon, we now provide more details on the sensitivity of other hyperparameters.
> Generally speaking, the results show that **our method is robust to most of the concerned hyperparameters, and its performance is close to the optimum when they are set within a reasonable range**.
>
> **RANSAC parameters for plane fitting.**
> During plane fitting via RANSAC, a point is considered an inlier if: (1) the angle between its normal and the normal hypothesis is less than $r_{n}$ AND (2) the distance from the point to the plane is less than $r_d$.
> As listed below, we report the performance of *NeuralPlane* under varying RANSAC parameters, including those adopted by other methods, as indicated in the last three rows.
> Thanks to the plane-biased scene geometry and the earlier scene decomposition based on learned coplanarity features, we find our method robust to the RANSAC parameters.
>
> | $(r_n$, $r_d)$                           | Chamfer $\downarrow$ | F-score $\uparrow$ | RI $\uparrow$ | VOI $\downarrow$ | SC $\uparrow$ |
> | :--------------------------------------- | :------------------- | :----------------- | :------------ | :--------------- | ------------- |
> | $($10$^{\circ}$, 2cm$)$                      | 4.64                 | 71.1               | 0.941$\text{\quad}$         | 2.61             | 0.297$\text{\quad}$         |
> | $($10$^{\circ}$, 5cm$)$                      | 4.57                 | 71.3               | 0.954         | 2.26             | 0.375         |
> | Our implementation: $($20$^{\circ}$, 8cm$)\text{\quad}$  | 4.59                 | 71.2               | 0.955         | 2.25             | 0.376         |
> | PlanarRecon: $($30$^{\circ}$, 25cm$)$        | 4.78                 | 70.3               | 0.953         | 2.26             | 0.378         |
> | AirPlanes: $($36.9$^{\circ}$, 30cm$)$        | 4.76                 | 70.5               | 0.951         | 2.26             | 0.382         |
> | PlanarNeRF: $($45.6$^{\circ}$, 35cm$)$                   | 4.73                 | 70.1               | 0.950         | 2.28             | 0.386         |
>
> **Pushing thresholds.**
> We then analyze the impact of different pushing thresholds, as proposed in *Eq. 7*.
> Results reported below show that small pushing thresholds are preferable, as they define a more geometrically strict coplanarity condition.
> However, when the thresholds become too strict, the coplanarity features tend to be excessively discriminative, resulting in over-segmentation and thus a decline in segmentation performance.
>
> | $(t_n$, $t_o)$                                | Chamfer $\downarrow$ | F-score $\uparrow$ | RI $\uparrow$ | VOI $\downarrow$ | SC $\uparrow$ |
> | :-------------------------------------------- | :------------------- | :----------------- | :------------ | :--------------- | ------------- |
> | $($cos10$^{\circ}$, 5cm$)$                      | 4.56                 | 71.3               | 0.949$\text{\quad}$         | 2.33             | 0.368$\text{\quad}$         |
> | Our implementation: $($cos10$^{\circ}$, 8cm$)\text{\quad}$ | 4.59                 | 71.2               | 0.955         | 2.25             | 0.376         |
> | $($cos20$^{\circ}$, 8cm$)$                      | 4.66                 | 70.2               | 0.950         | 2.31             | 0.370         |
> | $($cos30$^{\circ}$, 25cm$)$                     | 5.10                 | 68.9               | 0.952         | 2.32             | 0.358         |
> | $($cos60$^{\circ}$, 50cm$)$                     | 5.12                 | 68.3               | 0.950         | 2.37             | 0.365         |

---

> ### Author Response · Authors · 2024-11-20
> **2. Robustness to hyperparameters (2/2)**
>
> **Push margin.**
> As shown below, we empirically observe that separating a negative pair as far apart as possible (i.e., with a large push margin) can lead to faster convergence and improved performance.
> Due to a few local planar primitives that are not fully initialized at the beginning, we set the push margin $m$ to 1.5 for the first 1k iterations,
> and then increase it to 2.0, which represents the maximum distance in a unit hypersphere.
>
> | The push margin $m$                                          | Chamfer $\downarrow$ | F-score $\uparrow$ | RI $\uparrow$ | VOI $\downarrow$ | SC $\uparrow$ |
> | ------------------------------------------ | :------------------- | :----------------- | :------------ | :--------------- | ------------- |
> | 1.0                                        | 5.03                 | 69.1               | 0.948         | 2.32             | 0.373         |
> | 1.5                                        | 4.89                 | 70.0               | 0.953         | 2.29             | 0.366         |
> | 2.0                                        | 4.77                 | 70.5               | 0.950         | 2.25             | 0.376         |
> | 1.5 $\rightarrow$ 2.0 (Our implementation)$\text{\quad}$ | 4.59                 | 71.2               | 0.955$\text{\quad}$         | 2.25             | 0.376$\text{\quad}$         |
>
>
> **Loss balancing parameters.**
> The overall training objective is defined in *Eq. 9*, where three _intra-primitive_ loss terms $\mathcal{L}\_{\text{normal}}$, $\mathcal{L}\_{\text{p-depth}}$ and $\mathcal{L}\_{\text{pull}}$ are accordingly combined with balancing parameters $\lambda_1$, $\lambda_2$, and $\lambda_3$.
> Following DS-NeRF, we set $\lambda_2$ to $0.1$ without conducting further ablations.
> We now report the performance with different $\lambda_1$ and $\lambda_3$ respectively, since they correspond to two independent sets of training parameters.
> Results below show that $\lambda_1$ being too large could do harm to the geometry, while a smaller value may require more iterations for convergence.
>
> | $\lambda_1$: balancing parameter for $\mathcal{L}_{\text{normal}}\text{\quad}$ | Chamfer $\downarrow$ | F-score $\uparrow$ | RI $\uparrow$ | VOI $\downarrow$ | SC $\uparrow$ |
> | :------------------------------------------------------------------ | :------------------- | :----------------- | :------------ | :--------------- | ------------- |
> | 0.001                                                               | 5.19                 | 68.0               | 0.951$\text{\quad}$         | 2.31             | 0.372$\text{\quad}$         |
> | 0.01 (Our implementation)                                            | 4.59                 | 71.2               | 0.955         | 2.25             | 0.376         |
> | 0.05                                                                | 4.94                 | 69.7               | 0.948         | 2.32             | 0.370         |
> | 0.10                                                                | 5.78                 | 64.6               | 0.941         | 2.50             | 0.345         |
> | 1.00                                                                | 10.14                | 40.5               | 0.925         | 3.15             | 0.292         |
>
>
>
> Moreover, the results listed below indicate that $\lambda_3$=0.5 is an effective choice for balancing the pulling and pushing forces.
>
> | $\lambda_3$: balancing parameter for $\mathcal{L}_{\text{pull}}$ $\text{\quad}$ | Chamfer $\downarrow$ | F-score $\uparrow$ | RI $\uparrow$ | VOI $\downarrow$ | SC $\uparrow$ |
> | :---------------------------------------------------------------- | :------------------- | :----------------- | :------------ | :--------------- | ------------- |
> | 0.1                                                               | 4.99                 | 69.9               | 0.952$\text{\quad}$         | 2.29             | 0.365$\text{\quad}$         |
> | 0.5 (Our implementation)                                          | 4.59                 | 71.2               | 0.955         | 2.25             | 0.376         |
> | 1.0                                                               | 4.67                 | 70.2               | 0.951         | 2.29             | 0.373         |
> | 2.0                                                               | 5.08                 | 68.3               | 0.948         | 2.37             | 0.361         |

---

> ### Author Response · Authors · 2024-11-20
> **3. Out-of-domain experiment**
>
> Finally, following the constructive suggestion (**W1**) of the reviewer FCSv, we conducted an out-of-domain test on two small outdoor scenes.
> Two outdoor scenes from the training split of the [Niantic MapFree dataset](https://research.nianticlabs.com/mapfree-reloc-benchmark) were select for the experiment.
> All settings used in this experiment are kept consistent with those presented in our original submission.
> We also attempted to draw a comparison with baseline methods described in the main text.
>
> We find that *PlanarRecon* can not be successfully executed on both scenes, and the geometry+RANSAC baselines that employ learning-based MVS reconstruction methods (e.g., *SimpleRecon* and *DoubleTake*), fail to yield meaningful results.
> When compared to geometry+RANSAC baselines that employ neural implicit surface reconstruction methods (e.g., *NeuRIS* and *MonoSDF*), **our method produces reconstructions that are more complete and clean**.
>
> Since no ground-truth reconstruction is available, we would like to invite the reviewers to **check our updated supplementary material for qualitative evaluation**.

---

### Comment · Area_Chair_ZuLq · 2024-11-25
**Please read the rebuttal and reply**

Dear Reviewers,

Thanks again for serving for ICLR, the discussion period between authors and reviewers is approaching (November 27 at 11:59pm AoE), please read the rebuttal and ask questions if you have any. Your timely response is important and highly appreciated.

Thanks,

AC

---

### Meta-Review · Area_Chair_ZuLq · 2024-12-18

**Metareview:**

This paper proposes to method for multi-view 3D plane reconstruction. The main idea is to utilize foundational models to provide prior (normal, segmentation, etc) and then employ neural fields to learn a plane field that is aware of both geometry and scene semantics. The propose method presents an unified framework for both geometry and semantics, as well as a way to exploit strong prior from other models. During rebuttal, the major weakness raised by the reviewers is the complexity of the proposed method, which includes multiple modules with a handful of hyperparameters to tune. Other weaknesses include low efficiency due to the neural fields, missing related works, dependency of foundational models and paper presentation. The authors have actively addressed the concerns and provide supporting materials. After rebuttal, all reviewers suggest to accept the paper.

**Additional Comments On Reviewer Discussion:**

During rebuttal, the weakness raised by the reviewers are
- the complexity of the proposed method, which includes multiple modules with a handful of hyperparameters to tune.
- low efficiency due to the neural fields
- missing related works
- dependency of foundational models and paper presentation.

The authors have actively addressed the concerns, including testing their method for outdoor scenes, conducting ablation studies on hyperparameters, etc. Most of the concerns of reviewers are addressed by the authors' responses.

---

### Decision · Program_Chairs · 2025-01-22

Accept (Oral)